# Stomach-brain synchrony reveals a novel, delayed-connectivity resting-state network in humans

Ignacio Rebollo[1]*, Anne-Dominique Devauchelle[1,2], Benoît Béranger[3], Catherine Tallon-Baudry[1]

[1]Laboratoire de neurosciences cognitives, Département d'études cognitives, École normale supérieure, INSERM, PSL Research University, Paris, France; [2]Fondation Campus Biotech Geneva, Geneva, Switzerland; [3]Centre de NeuroImagerie de Recherche, Institut du Cerveau et de la Moelle épinière - ICM, Paris, France

**Abstract** Resting-state networks offer a unique window into the brain's functional architecture, but their characterization remains limited to instantaneous connectivity thus far. Here, we describe a novel resting-state network based on the delayed connectivity between the brain and the slow electrical rhythm (0.05 Hz) generated in the stomach. The gastric network cuts across classical resting-state networks with partial overlap with autonomic regulation areas. This network is composed of regions with convergent functional properties involved in mapping bodily space through touch, action or vision, as well as mapping external space in bodily coordinates. The network is characterized by a precise temporal sequence of activations within a gastric cycle, beginning with somato-motor cortices and ending with the extrastriate body area and dorsal precuneus. Our results demonstrate that canonical resting-state networks based on instantaneous connectivity represent only one of the possible partitions of the brain into coherent networks based on temporal dynamics.
DOI: https://doi.org/10.7554/eLife.33321.001

*For correspondence:
ignarebo@gmail.com

Competing interests: The authors declare that no competing interests exist.

## Introduction

The parsing of the brain into resting-state networks (RSNs) has been widely exploited to study the brain's functional architecture in health and disease (*Fox and Raichle, 2007*). With long-time scales, RSNs closely match the anatomical backbone of the brain (*van den Heuvel et al., 2009*; *Honey et al., 2009*; *Shen et al., 2015*). With short-time scales (~10–100 s), spontaneous brain activity is characterized by the emergence and dissolution of network patterns encompassing and extending classical RSN topologies (*Ponce-Alvarez et al., 2015*; *Shine et al., 2016*) with rich temporal trajectories (*Mitra et al., 2015*). Temporal trajectories indicate the existence of delays between regions, whereas the methods most often used to parse brain activity into functional networks (seed-based correlation and independent component analysis) make the implicit assumption that RSNs are characterized by instantaneous or zero delay connectivity. Therefore, we analyzed delayed connectivity in resting-state BOLD signals using techniques widely used in electrophysiological studies of large-scale brain dynamics (*Lachaux et al., 1999*) that quantify the stability of temporal delays between time series.

More specifically, we studied the delayed coupling between resting-state brain activity and a visceral organ, the stomach. The stomach continuously produces a slow electrical rhythm (0.05 Hz, one cycle every 20 s) that can be non-invasively measured (electrogastrogram, EGG [*Koch and Stern, 2004*]). The gastric basal rhythm is continuously (*Bozler, 1945*) and intrinsically (*Suzuki et al., 1986*) generated in the stomach wall by a network of specialized cells, the interstitial cells of Cajal

**eLife digest** The brain is always active. Even when it is not receiving sensory input, it generates its own spontaneous activity. This activity shapes how we interpret future sensory signals and creates our inner mental world. Moreover, this spontaneous activity is not random. When a healthy volunteer lies inside a brain scanner without performing any task, his or her brain shows predictable patterns of activity. Specific groups of brain regions – often with related roles – become active at the same time as one another. Each set of regions is referred to as a resting state network.

Of course, the brain does not operate in isolation from the rest of the body. Our internal organs continuously send signals to the brain via the spinal cord and cranial nerves. Specialized cells in the stomach wall in particular produce a slow rhythmic pattern of electrical activity. Known as the gastric rhythm, this activity helps ensure that the stomach muscles contract at the correct speed for digestion. But the stomach also produces this rhythm even when empty, suggesting that it has other roles too.

To find out what these might be, Rebollo et al. placed electrodes on the abdomen of healthy volunteers lying inside brain scanners. By examining the volunteers' spontaneous brain activity, Rebollo et al. identified a new resting state network that is active in synchrony with the gastric rhythm. The regions within this so-called gastric network are not active at the same time as each other, but instead become active in a specific sequence that is repeated at each gastric cycle. Many of the regions within the gastric network belong to other resting state networks too. Some of the regions help regulate automatic bodily functions such as heart rate, while others process information about the body's position in space.

The existence of the gastric network suggests a link between the automatic regulation of processes such as digestion, and spontaneous brain activity. Future studies could examine whether this link impacts perception and cognition, and whether this link plays a role in disorders where the connection between the digestive system and the brain appears to be altered.

DOI: https://doi.org/10.7554/eLife.33321.002

(*Sanders et al., 2014*), which form synapse-like connections not only with gastric smooth muscle but also with afferent sensory neurons (*Powley and Phillips, 2011*). The stomach is an interesting candidate for large-scale brain coordination for several reasons. First, visceral inputs can reach a number of cortical targets (*Critchley and Harrison, 2013*; *Park and Tallon-Baudry, 2014*). Second, gastric frequency (~0.05 Hz) falls within the range of BOLD fluctuations that are used to define RSNs and that are free from known cardiac and respiratory artifacts (*Glerean et al., 2012*). Finally, the amplitude of alpha rhythm, the dominant rhythm in the human brain at rest, depends on the phase of gastric rhythm (*Richter et al., 2017*).

We simultaneously recorded brain activity with fMRI and stomach activity with EGG (*Figure 1a*) in 30 human participants at rest with open eyes. We then determined the regions in which spontaneous fluctuations in brain activity were phase synchronized with gastric basal rhythm; we refer to these regions as the gastric network.

## Results

### EGG-BOLD phase coupling defines the gastric network

We first determined gastric frequency (*Figure 1b*) in each participant as the frequency of the largest spectral peak within the normogastric range (0.033–0.066 Hz). The mean EGG peak frequency across the 30 participants was 0.047 Hz (±SD 0.003, range 0.041–0.053). EGG peak frequency measured inside and outside the scanner did not differ (EGG outside the scanner measured in 29 of the 30 participants, mean 0.046 Hz ± SD 0.006; two-sided paired t-test, $t(28)=0.35$, $p=0.725$ Bayes Factor <0.001, indicating decisive evidence for the null hypothesis).

In each participant and at each voxel, we quantified the degree of phase synchrony between the EGG signal and BOLD time series filtered around gastric frequency (*Figure 1c*). We computed the phase-locking value (PLV) (*Lachaux et al., 1999*), a measure widely used in electrophysiology that varies between zero when two time series show no consistent phase relationship (*Figure 1c*, bottom

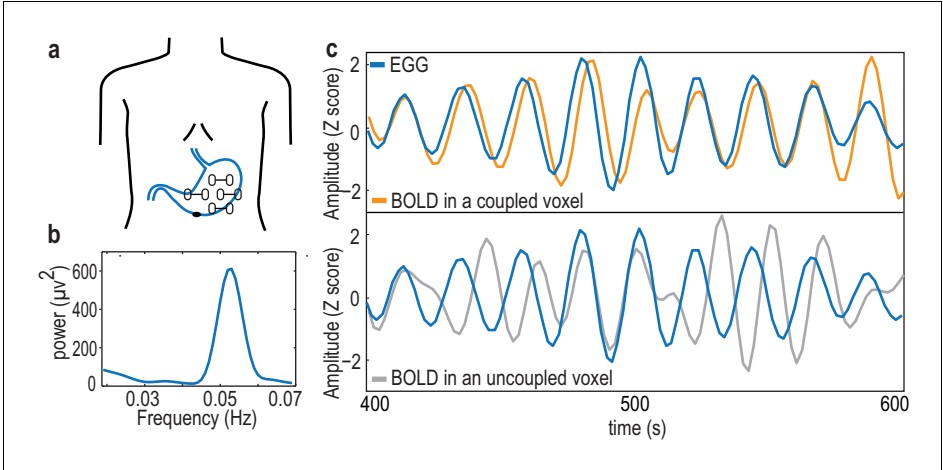

**Figure 1.** The electrogastrogram (EGG) and gastric-BOLD coupling. (a) Bipolar electrode montage used to record the EGG. (b) Example of an EGG spectrum in a single participant, with the typical spectral signature in the normogastric frequency range (0.033–0.066 Hz). (c) Example of a 200 s time series of the EGG and BOLD signal filtered at the gastric peak frequency in an EGG-coupled voxel (top), characterized by a stable, non-zero phase relationship between EGG and BOLD, resulting in a high phase-locking value, and in an EGG-uncoupled voxel (bottom) in which no stable phase relationship between EGG and BOLD can be observed.
DOI: https://doi.org/10.7554/eLife.33321.003

panel) and one when two time series have a consistent phase relationship over time (*Figure 1c*, upper panel). PLV has three important properties: PLV is high for any lag between the time series as long as this lag is constant over time, PLV is independent of signal amplitude, and PLV gives no indication on the directionality of interactions between the two time series. In each participant and at each voxel, we estimated the PLV that could be expected by chance from EGG signals that were shifted with respect to the BOLD time series. The empirical PLVs were then compared with chance-level PLVs using a cluster-based statistical procedure that intrinsically corrects for multiple comparisons (*Maris and Oostenveld, 2007*). Significant phase coupling between the EGG and resting-state BOLD time series occurred in twelve nodes (voxel threshold p<0.01, two-sided paired t-test between observed and chance PLV; cluster threshold corrected for multiple comparisons, Monte-Carlo p<0.05). Exact p-values are reported for each cluster in *Table 1*.

The gastric network (*Table 1*, *Figure 2a*) comprises the right primary somatosensory cortex (SIr), bilateral secondary somatosensory cortices (SII), medial wall motor regions (MWM), comprising the caudate cingulate motor zone (CCZ), posterior rostral cingulate motor zone (RCZp), and right supplementary motor area (SMA), a region of the right occipito-temporal cortex overlapping the extrastriate body area (EBA), as well as nodes in the posterior cingulate sulcus (pCS), dorsal precuneus (dPrec), occipital cortex (ventral and dorsal portions, vOcc and dOcc), retrosplenial cortex (RSC), and superior parieto-occipital sulcus (sPOS). Estimating chance-level PLV by computing gastric-BOLD coupling between the BOLD signal of one participant with the EGG of the other 29 participants resulted in a qualitatively similar network, with coupling occurring either in the same or neighboring voxels (Supplemental *Figure 2*). The average shared variance between the EGG and BOLD signals across participants, as estimated from squared coherence, ranged from 12% in the left anterior dorsal precuneus to 16.9% in the posterior cingulate sulcus (*Table 1*).

An analysis of covariance across nodes did not reveal significant links between gastric-BOLD coupling strength (defined as the difference between empirical and chance PLV) and gender (F(1, 28) =1.02, p=0.46), body mass index (BMI) (F(1, 28)=1.3, p=0.3) or trait anxiety scores (F(1, 28)=1.02, p=0.47. Statistics (including Bayes Factor) per node are reported in *Table 2*. Note that there is less variation in BMI in our sample than in the general population since all participants had a BMI smaller than 25.

**Table 1.** Description of the 12 nodes showing larger-than-chance gastric-BOLD phase coupling.
AAL: Automated Anatomical labeling (*Tzourio-Mazoyer et al., 2002*). MNI: Montreal Neurological Institute.

| Node name | Monte-Carlo p | Sum (t) | Cluster volume in mm³ | Label AAL | mm³ | % Area in Cluster | Max t | MNI of max t X | Y | Z | EGG-BOLD shared Variance ± sem |
|---|---|---|---|---|---|---|---|---|---|---|---|
| Primary Somatosensory Right (SIr) | 0.0049 | 216.5 | 180 | Postcentral R | 171 | 8.8 | 4.8 | 45 | −28 | 46 | 12.5% ± 2.0 |
| Secondary Somatosensory Right (SIIr) | 0.0131 | 146.1 | 120 | Rolandic Oper R | 12 | 1.8 | 4.2 | 54 | −22 | 13 | 15.7% ± 2.2 |
| | | | | Heschl R | 6 | 4.7 | 3.9 | 51 | −19 | 10 | |
| | | | | Temporal Sup R | 99 | 6 | 5.3 | 57 | −25 | 13 | |
| Secondary Somatosensory Left (SIIl) | 0.0094 | 167.0 | 138 | Postcentral L | 39 | 2.0 | 4.5 | −66 | −22 | 22 | 16.6% ± 2.2 |
| | | | | SupraMarginal L | 21 | 3.3 | 3.8 | −60 | −25 | 16 | |
| | | | | Heschl L | 15 | 13.1 | 3.9 | −54 | −16 | 7 | |
| | | | | Temporal Sup L | 60 | 5 | 5.2 | −54 | −19 | 7 | |
| Medial Wall Motor Regions (MWM) | 0.0036 | 269.5 | 228 | Supp Motor Area L | 30 | 2.7 | 3.8 | 0 | −13 | 49 | 15.8% ± 2.1 |
| | | | | Supp Motor Area R | 54 | 4.5 | 4.8 | 9 | -4 | 52 | |
| | | | | Cingulum Mid L | 66 | 6.7 | 4.9 | 0 | −22 | 43 | |
| | | | | Cingulum Mid R | 75 | 6.7 | 4.7 | 6 | −10 | 43 | |
| Posterior Cingulate Sulcus (pCS) | 0.0061 | 196.7 | 171 | Cingulum Mid L | 39 | 3.9 | 4.2 | -9 | −37 | 52 | 16.9% ± 2.3 |
| | | | | Cingulum Mid R | 24 | 2.1 | 4.5 | 6 | −37 | 49 | |
| | | | | Precuneus L | 21 | 1.2 | 3.9 | 0 | −37 | 55 | |
| | | | | Precuneus R | 39 | 2.3 | 3.8 | 6 | −40 | 52 | |
| | | | | Paracentral Lobule R | 39 | 9.1 | 4.1 | 15 | −40 | 55 | |
| Dorsal Precuneus (dPrec) | 0.0071 | 186.6 | 156 | Precuneus L | 42 | 2.3 | 4.9 | -3 | −67 | 61 | 12.7% ± 1.9 |
| | | | | Precuneus R | 87 | 5.2 | 4.4 | 3 | −64 | 64 | |
| Dorsal Precuneus Left Anterior (ladPrec) | 0.0178 | 125.5 | 108 | Precuneus L | 57 | 3.2 | 4.1 | -6 | −55 | 73 | 12.0% ± 2.0 |
| | | | | Paracentral Lobule L | 48 | 7.0 | 4.3 | -6 | −34 | 76 | |
| Occipital Ventral (vOcc) | 0.0017 | 374.2 | 321 | Calcarine L | 18 | 1.6 | 4.1 | 0 | −64 | 10 | 16.6% ± 2.6 |
| | | | | Lingual L | 66 | 6 | 4.4 | 0 | −67 | 7 | |
| | | | | Lingual R | 108 | 9 | 4.4 | 6 | −70 | -5 | |
| | | | | Cerebellum 6 R | 21 | 2.3 | 4.3 | 12 | −70 | −14 | |
| | | | | Vermis 4 5 | 51 | 15.0 | 4.6 | 0 | −61 | -2 | |
| | | | | Vermis 6 | 9 | 4.8 | 3.7 | 3 | −70 | -8 | |
| Occipital Dorsal (dOcc) | 0.0034 | 285.1 | 243 | Calcarine L | 21 | 1.8 | 4 | 3 | −76 | 16 | 16.7% ± 2.7 |
| | | | | Cuneus L | 138 | 17.7 | 4.5 | -3 | −85 | 31 | |
| | | | | Cuneus R | 72 | 9.9 | 4.9 | 6 | −76 | 28 | |
| Extrastriate Body Area Right (EBA) | 0.0103 | 163.3 | 138 | Occipital Inf R | 45 | 8.9 | 4.5 | 48 | −76 | −14 | 14.3% ± 2.0 |
| | | | | Temporal Inf R | 27 | 1.5 | 4.4 | 57 | −70 | −11 | |
| | | | | Cerebellum Crus1 R | 33 | 2.4 | 4.8 | 45 | −82 | −26 | |
| Superior Parieto-Occipital Sulcus (sPOS) | 0.0245 | 107.1 | 87 | Cuneus R | 21 | 2.9 | 4.6 | 21 | −76 | 43 | 12.5% ± 1.8 |
| | | | | Occipital Sup R | 54 | 7.5 | 5.1 | 24 | −79 | 43 | |

*Table 1 continued on next page*

Table 1 continued

| Node name | Monte-Carlo p | Sum (t) | Cluster volume in mm³ | Label AAL | mm³ | % Area in Cluster | Max t | X | Y | Z | EGG-BOLD shared Variance ± sem |
|---|---|---|---|---|---|---|---|---|---|---|---|
| | | | | | | | | MNI of max t | | | |
| Retrosplenial Cortex (RSC) | 0.0084 | 175.6 | 147 | Cingulum Post L | 3 | 1.3 | 3.5 | -6 | −43 | 10 | 15.4% ±2.1 |
| | | | | Cingulum Post R | 9 | 5.3 | 3.7 | 6 | −40 | 7 | |
| | | | | Lingual R | 18 | 1.5 | 4.1 | 9 | −37 | -2 | |
| | | | | Precuneus L | 36 | 2.0 | 4.4 | -6 | −49 | 7 | |

DOI: https://doi.org/10.7554/eLife.33321.004

## Controls: gastric frequency specificity, false-positive rate, and head micromovements

To assess the robustness of the gastric network, we ran several controls. First, we verified that EGG-BOLD coupling was specific to gastric frequency. We filtered both EGG and BOLD time series at frequencies that were slightly offset from the peak gastric frequency of each participant and recomputed cluster statistics. Summary statistics (sum of the absolute t-values resulting from the paired t-test between empirical and chance-level PLV at each voxel, either summed across the whole brain or within the gastric network) decreased when shifting below or above the gastric peak frequency (*Figure 2b*). This result indicates that the gastric network corresponds to BOLD fluctuations specifically occurring at gastric frequency.

Second, we estimated the likelihood of false positives with our statistical procedure. We randomly sampled surrogate datasets in which a random time shift was applied to the EGG of each participant a thousand times. Next, we tested whether any of those 1000 combinations would generate summary statistics as large as the original data when compared with the chance-level estimate we used to determine significantly coupled regions at the group level (*Figure 2c*). This result was never observed, indicating that the probability of our results being a false positive is less than 0.001.

Third, we verified that gastric-BOLD coupling strength was unrelated to BOLD power at gastric frequency. We computed the correlation between BOLD power at gastric frequency and coupling strength for each participant and voxel, and found the two measures to be unrelated (Fisher z-transformed Pearson correlation coefficients tested against zero, t(29)=1.19, p=0.24; Bayes factor <0.001, indicating decisive evidence for the absence of a link between coupling strength across the brain and BOLD power at gastric frequency).

Finally, we investigated whether submillimeter head movements might have influenced the results. We defined voxel motion susceptibility as the regression coefficient of head movement (*Power et al., 2012*) from the BOLD time series. Coupling strength and voxel motion were unrelated (Fisher z-transformed Pearson correlation coefficients tested against zero, t(29)=-0.34, p=0.73; Bayes factor <0.001, indicating decisive evidence for the absence of a link between coupling strength and head movement). Stomach contractions might also lead to small head movements that could be phase locked to gastric rhythm. Although gastric rhythm is continuously produced even during fasting, it is larger during stomach contractions. Thus, we tested whether the effects we found were due to differences in EGG power (or frequency) across participants. We found no link between coupling strength in the 12 nodes and EGG power (ANCOVA, F(1, 28)=0.9, p=0.51; all Bayes factor <0.33, indicating substantial evidence for the null hypothesis) nor between coupling strength and EGG peak frequency (ANCOVA, F(1, 28)=1.6, p=0.17; Bayes Factor <0.33, indicating substantial evidence for the null hypothesis in 9 of 12 nodes; Bayes Factor <1.3 in the three remaining nodes, indicating anecdotal or no evidence).

The gastric network is thus specific to individual gastric peak frequency, is highly unlikely to be a chance finding, is not dependent on BOLD power, and is not linked to spurious effects of head movement on the BOLD signal.

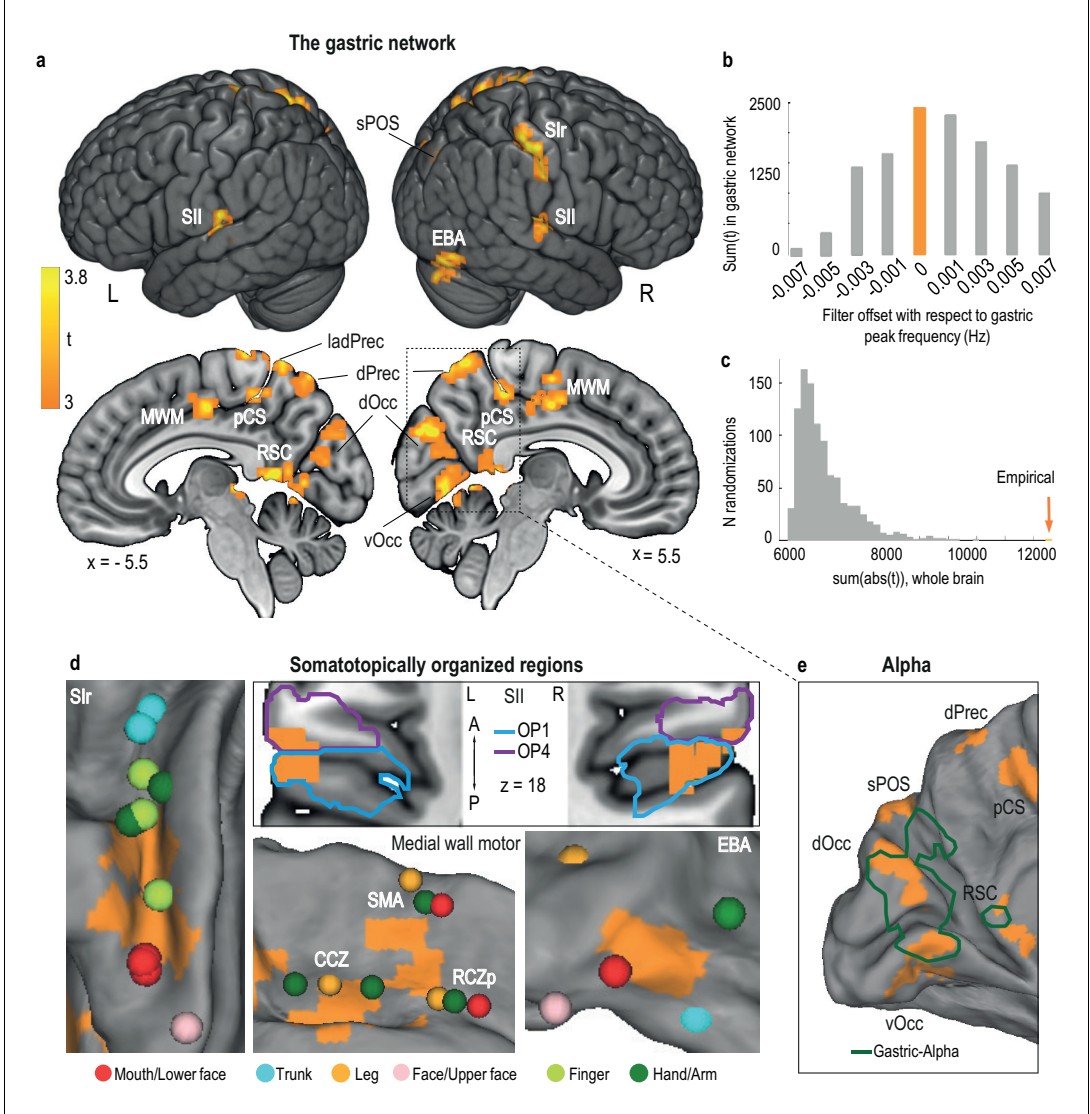

**Figure 2.** The gastric network. (**a**) Regions significantly phase synchronized to gastric rhythm (n = 30, voxel level threshold, p<0.01 two-sided; cluster level threshold, p<0.05, two-sided, intrinsically corrected for multiple comparisons). (**b**) Gastric-BOLD coupling is specific to gastric frequency. Summary statistics in the gastric network are maximal at the EGG peak frequency (orange) and decrease when offsetting the filter to higher or lower frequencies. (**c**) Summary statistics distribution under the null hypothesis from 1000 surrogate datasets in which the EGG signal was time-shifted with respect to the BOLD signal. The empirical finding (orange arrow) falls well outside the null distribution. (**d**) The gastric network (orange) comprises the following somatotopically organized regions: *primary somatosensory cortex* (*Panel SI*, with peak activations during stimulation of the trunk and hand (***Fabri et al., 2005***), finger (***Weibull et al., 2008***), face (***Kopietz et al., 2009***), and mouth, that is, teeth (***Trulsson et al., 2010***), lips and tongue (***Miyamoto et al., 2006***); *secondary somatosensory cortex* (*Panel SII*, cytoarchitectonic subdivisions of SII according to (***Scheperjans et al., 2008***); OP1, parietal operculum 1 and OP4, parietal operculum 4, presented on a horizontal slice at z = 18); *medial wall motor areas* (*Panel MWM*, with peak activations during movement (***Amiez and Petrides, 2014***) in the caudate cingulate zone (CCZ), posterior rostral cingulate zone (RCZp) and supplementary motor area (SMA)); and *extrastriate body area* (*Panel EBA* with peak activations during body part viewing (***Orlov et al., 2010***), note that because of the visualization on an inflated cortex, the extension of the EBA node to the cerebellum is not visible). (**e**) Regions in which the alpha and gastric rhythms are coupled (green, modified from [***Richter et al., 2017***]). Abbreviations are the same as those in ***Table 1***.

DOI: https://doi.org/10.7554/eLife.33321.005

The following source data and figure supplement are available for figure 2:

**Source data 1.** csv file containing the sum of t in gastric network resulting from filtering the EGG and the BOLD signal with an offset with respect to gastric peaking frequency, used for *Figure 1b*.
DOI: https://doi.org/10.7554/eLife.33321.007
**Source data 2.** csv file containing the sum of abs(t) in the whole brain for each randomization of time-shift of the EGG, used for *Figure 1c*.
DOI: https://doi.org/10.7554/eLife.33321.008

*Figure 2 continued on next page*

*Figure 2 continued*

**Figure supplement 1.** The gastric-network when estimating chance-level PLV by computing gastric-BOLD coupling between the BOLD signal of one participant with the EGG of the other 29 participants.

DOI: https://doi.org/10.7554/eLife.33321.006

## The gastric network includes body maps associated with touch, action and vision

We then examined the areas comprising the gastric network in more detail. By definition, the gastric network is composed of regions with activity that co-fluctuates with gastric basal rhythm. Five nodes of the gastric network also share a common functional feature, somatotopic organization, as detailed in *Figure 2d*.

The gastric network includes the following regions with a well-known body representation based on touch: the right primary somatosensory cortex in the hand and mouth region and bilateral secondary somatosensory cortices. We quantified the overlap between these gastric network nodes and known cytoarchitectonic subdivisions of the somatosensory cortices (*Geyer et al., 2000*; *Grefkes et al., 2001*). The gastric network mostly overlapped with area 1 (60.2% of the SIr node) and to a lesser extent, with area 2 (13.1%) and area 3b (9.9%). The SII nodes of the gastric network overlapped with the secondary somatosensory cortices and more precisely with the somatotopically organized subdivisions of the parietal operculum OP1 and OP4 (22). The right SII node mostly overlapped with area OP1 (35.2% of the node), while the left SII node overlapped with both OP1 (21.7%) and OP4 (14.9%). Additionally, both left and right SII nodes extended more ventrally to the temporal cortex.

The gastric network also includes three medial wall motor regions (CCZ, RCZp, and SMA) that reveal their somatotopic organization when participants are required to move specific body parts (*Amiez and Petrides, 2014*). Note that gastric-BOLD coupling also included a more posterior area in the cingulate sulcus (pCS). Finally, the gastric network overlapped with the EBA, a region of the lateral occipital cortex activated when participants view images of body parts (*Downing et al.,*

**Table 2.** Effects of demographical variables on coupling strength at each node.

Statistics are not corrected for multiple comparisons. Bayes factor smaller than 0.33 indicate evidence for the null hypothesis, Bayes factor larger than three indicate evidence for an effect

| | Gender | | | BMI | | | Stai A | | |
|---|---|---|---|---|---|---|---|---|---|
| | t | p | Bf | r | p | Bf | r | p | Bf |
| SI | 1.537 | 0.146 | 0.830 | 0.072 | 0.707 | 0.151 | 0.022 | 0.909 | 0.142 |
| SIIr | 0.476 | 0.642 | 0.375 | 0.187 | 0.322 | 0.230 | −0.084 | 0.657 | 0.156 |
| SIIl | 2.542 | 0.023 | 3.455 | 0.270 | 0.150 | 0.396 | −0.272 | 0.146 | 0.402 |
| RCZp | 1.429 | 0.175 | 0.738 | 0.107 | 0.575 | 0.165 | −0.255 | 0.174 | 0.353 |
| pCS | 2.506 | 0.025 | 3.250 | 0.142 | 0.454 | 0.187 | −0.296 | 0.112 | 0.493 |
| dPrec | 1.265 | 0.227 | 0.628 | 0.035 | 0.853 | 0.144 | −0.243 | 0.196 | 0.324 |
| dPrec la | 1.289 | 0.218 | 0.642 | 0.224 | 0.234 | 0.285 | −0.215 | 0.254 | 0.269 |
| vOcc | 2.342 | 0.034 | 2.487 | 0.342 | 0.065 | 0.769 | 0.003 | 0.988 | 0.141 |
| dOcc | 2.075 | 0.057 | 1.656 | 0.337 | 0.069 | 0.729 | −0.068 | 0.723 | 0.150 |
| EBA | 1.882 | 0.081 | 1.267 | −0.064 | 0.735 | 0.150 | −0.250 | 0.183 | 0.341 |
| sPOS | 2.164 | 0.048 | 1.889 | 0.481 | 0.007 | 5.133 | −0.143 | 0.450 | 0.187 |
| RSC | 1.129 | 0.278 | 0.556 | 0.105 | 0.579 | 0.165 | −0.317 | 0.088 | 0.600 |

DOI: https://doi.org/10.7554/eLife.33321.009

The following source data available for Table 2:

**Source data 1.** csv with data per subject: gender, trait anxiety, BMI, and coupling strength at each node.

DOI: https://doi.org/10.7554/eLife.33321.010

*2001*) with a clear somatotopic organization (*Orlov et al., 2010*). The overlap between the gastric network and EBA occurred in the lower face region, which includes the mouth.

Thus, the gastric network overlaps with body maps classically associated with different modalities, including touch in somatosensory cortices, action in MWM and vision in the EBA.

## The gastric network includes regions involved in the generation of the alpha rhythm

Finally, we found gastric-BOLD coupling in the posterior bank of the parieto-occipital sulcus (dOcc and vOcc) and retrosplenial cortex. In a previous study using magneto-encephalography (*Richter et al., 2017*), the amplitude of the alpha rhythm in these regions was modulated by gastric phase (*Figure 2e*).

## Gastric-brain coupling in the right posterior insula

The insula is one region that receives visceral inputs (*Critchley and Harrison, 2013*; *Park and Tallon-Baudry, 2014*), but it did not appear to be significantly phase synchronized to the EGG using our whole-brain, statistically conservative procedure. Thus, we performed post-hoc region-of-interest analysis of the three insular subdivisions (anterior dorsal, anterior ventral, posterior [*Deen et al., 2011*]) in both hemispheres. Only the right posterior insula showed evidence for gastric-BOLD coupling across participants (empirical vs. chance-level PLV, paired t-test, two sided, t(29)=2.78, p=0.043, Bonferroni corrected; all other regions, p>0.21).

The use of statistical thresholds results in binary outputs. To get a finer grained picture, we computed effect sizes in the 6 insular subdivisions and in the 12 gastric network nodes (Cohen's d for the difference between empirical and chance PLV on the mean time series in each region of interest). Mean Cohen's d across gastric network nodes was 1.19 ± 0.21 STD, ranging from 0.80 in the dorsal occipital cortex to 1.62 in the right secondary somatosensory cortex. The right posterior insula had an effect size of 0.84, within the lower range of the gastric network. All other insula subdivisions displayed smaller effect sizes (right: dorsal anterior 0.61, ventral anterior 0.54; left: posterior 0.60, dorsal anterior 0.41, ventral anterior = 0.52). Thus, the right posterior insula does show evidence for coupling with the stomach, with an effect size comparable to that of the weakest nodes of the gastric network, provided signal-to-noise ratio is first increased by averaging within a region of interest.

## Partial overlap between gastric network and autonomic networks

Is the gastric network specific to the stomach, or is it also linked to other organs such as the heart? We determined brain regions (FWE corrected p<0.05; *Figure 3A*) fluctuating with high- and low-frequency heart rate variability, that represent parasympathetic and a mixture of sympathetic and parasympathetic outputs, respectively. Of the gastric network, 30% was also related to heart rate variability, mostly in medial motor regions and in the posterior cingulate cluster (low-frequency heart rate variability), and, to a lesser extent, in the dorsal occipital cluster (high-frequency heart rate variability). Because we did not record any measure that isolates sympathetic output, we additionally analyzed the overlap between the gastric network and known sympathetic areas (*Beissner et al., 2013*). This overlap was very limited (34 voxels, 4.7% of the gastric network) and confined to SIr and the anterior parts of medial motor regions (*Figure 3—figure supplement 1*).

We also determined brain regions that correlate with pupil diameter (n = 20 due to data loss or artefacts; *Figure 3B*). The strongest correlations were found in occipital regions, somato-motor cortices and medial wall motor regions. 17% of the gastric network (SI, SIIr, MWM and EBA) overlaps with regions correlating with pupil diameter. Shared variance between pupil diameter and EGG, estimated from squared coherence, was 9.7 ± 2.5%. Coupling strength averaged across SI, SIIr, MWM and EBA did not correlate with shared pupil-EGG variance (mean r = 0.05, p=0.82, BF = 0.17 which indicates substantial evidence for the null hypothesis).

## Temporal sequence within a gastric cycle and delayed connectivity between the nodes of the gastric network

In the different nodes of the gastric network, gastric-brain coupling occurred with different phase delays with respect to the gastric cycle. We analyzed between-participant phase-delay consistency and found temporal delays of ~3.3 s between the earliest nodes (somatosensory cortices) and latest

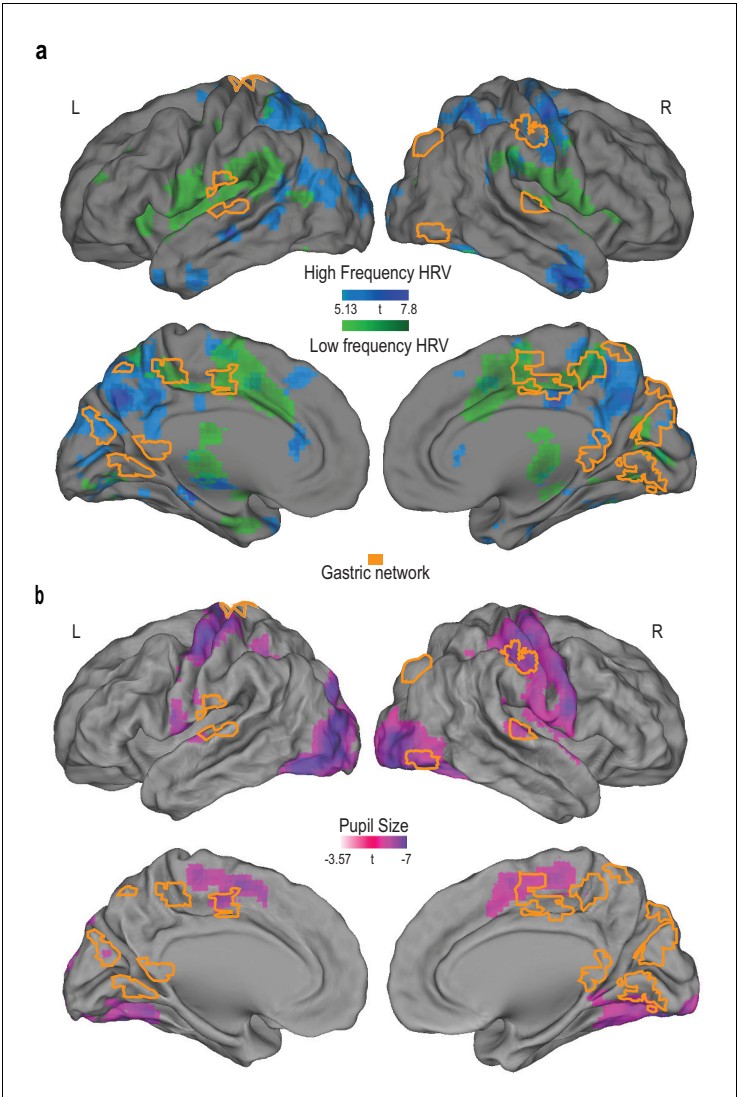

**Figure 3.** Overlap between the gastric network and regions correlating with heart rate variability and pupil size. (A) Random effects analysis across participants (N = 30), for the main effects of high- (blue) and low (green)-frequency heart rate variability power fluctuations presented on an inflated brain ($p_{FWE} < 0.05$ and 30 voxels). The gastric network is represented in orange. (B) Random effects analysis across participants (N = 20), for the main effect of pupil diameter (pink; threshold p<0.001 and 30 voxels). At this threshold, only negative correlations were observed.

DOI: https://doi.org/10.7554/eLife.33321.011

The following source data and figure supplement are available for figure 3:

**Source data 1.** Csv with peak coordinates for group results on high frequency heart rate variability.
DOI: https://doi.org/10.7554/eLife.33321.013
**Source data 2.** Csv with peak coordinates for group results on low frequency heart rate variability.
DOI: https://doi.org/10.7554/eLife.33321.014
**Source data 3.** Csv with peak coordinates for group results on pupil diameter.
DOI: https://doi.org/10.7554/eLife.33321.015
**Figure supplement 1.** Overlap between the gastric network and meta-analytic sympathetic and parasympathetic regions.
DOI: https://doi.org/10.7554/eLife.33321.012

nodes (dorsal precuneus and EBA) of the gastric network (*Figure 4a,b*). The delay in the right posterior insula was in the range of the earliest nodes of the gastric network (*Figure 4a*). The Watson-Williams test for circular data confirmed that different nodes of the gastric network were coupled to the gastric rhythm with different phase delays (F(11, 29)=5.22, p<$10^{-6}$), indicating a precise temporal sequence of activations within each gastric cycle.

Thus, each node of the gastric network appears to be characterized by a specific temporal delay with respect to gastric phase. These temporal delays were accompanied by delayed functional connectivity (FC) between the nodes of the gastric network.

We first illustrated this point with an example in a single participant (*Figure 4c*), with two 200 s time series of the gastric network (MWM and EBA). The two time series systematically co-varied with a temporal delay. The existence of temporal delays between the nodes of the gastric network is one of the reasons why the gastric network could not be observed in prior studies. Indeed, fMRI RSN studies are typically based on measures of instantaneous FC, such as shared variance estimated from the squared Pearson correlation coefficient, which does not detect the temporally delayed interactions revealed here. These measures differ from delayed FC measures based on the consistency of phase delays over time, such as shared variance estimated from squared coherence. In the example illustrated in *Figure 4c*, instantaneous FC between the two time series is 56%, whereas delayed FC is 86%. If we advance the timing of the medial wall time series by 2 s, instantaneous FC increases to 86%. This finding shows that the difference between the two FC estimates is due to temporal delays only.

We then estimated both instantaneous and delayed FC between all nodes of the gastric network in all participants. Delayed FC between gastric nodes (mean 40.8% ± SD 8%, ranging from 26.5% between the right primary somatosensory cortex and RSC, up to 63.9% between the ventral and dorsal occipital cortices) was systematically larger (paired t-test, t(29)=9.02, p<$10^{-10}$) than instantaneous FC (mean 30.2% ± SD 11%, ranging from 9.3% between the dorsal precuneus and right SII, up to 61.2% between right and left SII). Next, we verified (*Figure 4d,e*) that two regions belonging to both the gastric network and the same RSN (i.e. two regions of the gastric network with little temporal delay, such as MWM and SIr) would display large values of both delayed and instantaneous FC, whereas two regions belonging to the gastric network but not to the same classical RSN (i.e. two regions of the gastric network with a large temporal delay, such as MWM and EBA) would show large delayed FC and small instantaneous FC. Thus, in contrast to classical RSNs, the gastric network appears to be characterized by between-node delayed connectivity.

## Slow temporal fluctuations in gastric-BOLD coupling are associated with changes in BOLD amplitude and occur simultaneously in all nodes

Thus far, we have identified a sequence of activation that occurs at each gastric cycle, which characterizes gastric-BOLD coupling. We then investigated whether slow temporal fluctuations in the strength of gastric-BOLD coupling were accompanied by fluctuations in BOLD amplitude. As illustrated in *Figure 5a*, we found that episodes of elevated gastric-BOLD synchronization corresponded to episodes of increased BOLD amplitude. Indeed, time-varying PLV and BOLD time series, computed in sliding time windows of 60 s (approximately three gastric cycles), were significantly correlated (Fisher z-transformed Pearson correlation coefficients t-tested against zero, Bonferroni corrected p<0.006 in all gastric nodes, mean r across nodes 0.18 STD =± 0.02, ranging from 0.15 in MWM to 0.22 in SIIl).

Next, we tested whether slow temporal fluctuations in gastric-BOLD synchronization occurred simultaneously or independently in the different nodes of the gastric network (*Figure 5b*). We computed the correlation between time-varying PLVs for all possible node pairs in each participant and found that at the group level, this correlation was significantly positive (Fisher z-transformed average Pearson correlation coefficients against zero, t(29)=9.22, p<$10^{-10}$, mean r = 0.129 ± STD 0.075, range across participants 0.02–0.35). To determine whether the overall pattern of synchronous fluctuations in gastric-BOLD coupling strength was driven by specific node pairs, we investigated correlations between node pairs. All node pairs but RSC-SIr, sPOS-RSC, sPOS-SIIr and sPOS-ladPrec showed a significant positive correlation at the group level (Fisher z-transformed Pearson correlation coefficients in all pairs tested against zero, Bonferroni corrected). The three nodes showing the largest covariation in time-varying PLV with the other nodes were dPrec, pCS and vOcc, and the three nodes showing the least covariation in time-varying PLV with the other nodes were RSC, SIr and

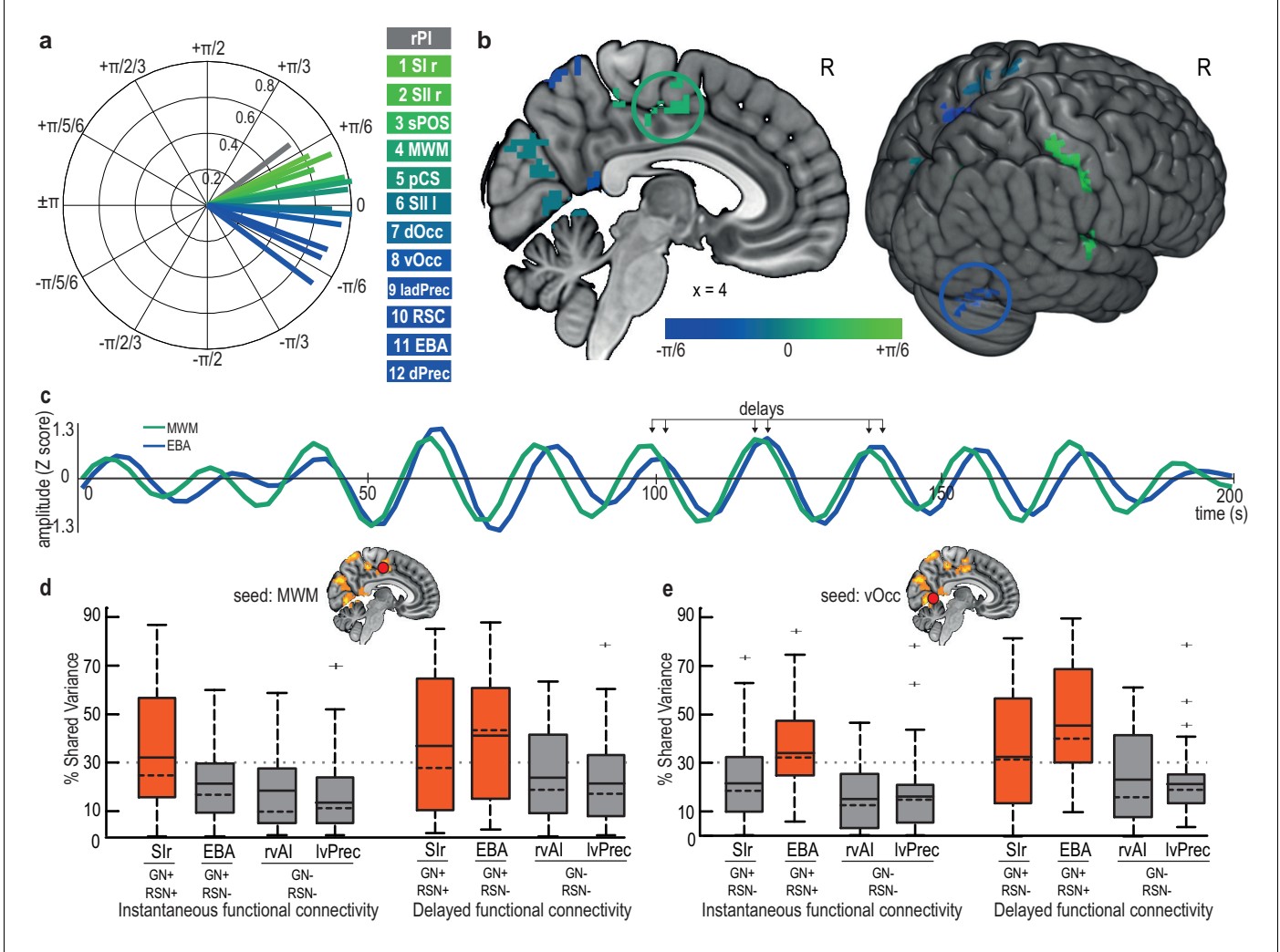

**Figure 4.** Early and late nodes of the gastric network are functionally connected but with a delay. (**a**) Phase-delay consistency of each node of the gastric network. Vector length quantifies the phase consistency across participants, and vector angle indicates phase advance (green) or phase lag (blue) within the gastric network. Temporal delays can reach up to ~3.3 s ($\pm\pi/6$) between the earliest nodes (primary and secondary somatosensory cortices) and the latest nodes (dorsal precuneus and EBA). The right posterior insula (gray) is earlier than any node of the gastric network. (**b**) Group-averaged phase delays for each cluster in the gastric network. The two circled regions (EBA, blue; MWM, green) are illustrated in C. (**c**) A 200 s BOLD time series in MWM and EBA in a single participant, showing phase consistency with delays. (**d**). Group-level functional connectivity across all participants using MWM as a seed, either instantaneous (left) or delayed (right), with regions belonging or not to the gastric network (GN+/-, defined by delayed connectivity) and with regions belonging or not to the same resting-state network (RSN+/-, defined by instantaneous connectivity). Boxes are colored red when the mean FC exceeds 30%. The boxplot presents the mean (full line), median (dashed line), first and third quartiles (box), and extrema (whiskers) excluding outliers (+, defined as exceeding 1.5 interquartile ranges above the third quartile). (**e**) Group-level functional connectivity across all participants using the vOcc as a seed. Abbreviations: EBA, extrastriate body area; MWM, medial wall motor regions; rSI, right primary somatosensory cortex; rVAI, right ventral anterior insula; lvPrec, left ventral precuneus.

DOI: https://doi.org/10.7554/eLife.33321.016

The following source data is available for figure 4:

**Source data 1.** csv containing the shared variance between MWM time series and SIr, EBA, rvAI and lvPrec time series of each participant using the squared pearson coefficient (columns 1 to 4), and the squared coherence coefficient (columns 5 to 8).
DOI: https://doi.org/10.7554/eLife.33321.017

**Source data 2.** csv containing the shared variance between vOcc time series and SIr, EBA, rvAI and lvPrec time series of each participant using the squared pearson coefficient (columns 1 to 4), and the squared coherence coefficient (columns 5 to 8).
DOI: https://doi.org/10.7554/eLife.33321.018

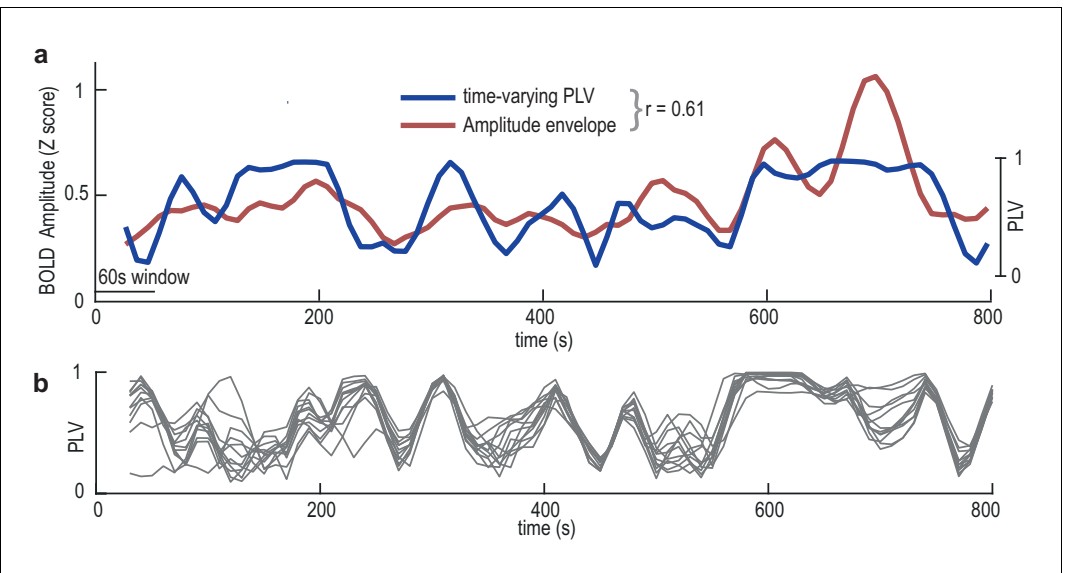

**Figure 5.** Dynamic fluctuations in EGG-BOLD coupling. (**a**) Data from the superior parieto-occipital sulcus in a single participant showing the correlation (r = 0.61) between the time-varying gastric-BOLD coupling (PLV, blue) and the amplitude envelope of the BOLD time series (red). (**b**) Time-varying PLV at each node of the gastric network in the same participant showing a strong correlation across nodes (average r across all possible node pairs 0.35).

DOI: https://doi.org/10.7554/eLife.33321.019

sPOS. Thus, slow temporal fluctuations in gastric-BOLD coupling are associated with changes in BOLD amplitude and occur simultaneously in all nodes.

## Discussion

Here, we reveal the existence of the gastric network, comprising brain regions with BOLD time series that are phase synchronized with gastric basal rhythm. Within the gastric network, approximately 15% of the BOLD variance is explained by gastric-BOLD phase synchrony. The gastric network cuts across classical RSNs and shows only partial overlap with autonomic control regions. A number of brain regions composing the gastric network have convergent functional properties involved in mapping bodily space through touch, action and vision. The network is characterized by a precise temporal sequence of activations within a gastric cycle, beginning with somato-motor cortices and ending with extra-striate body area and dorsal precuneus. This temporal sequence is accompanied by delayed functional connectivity between nodes of the gastric network, which explains why this RSN could not be identified with standard correlation methods that only capture instantaneous connectivity. Furthermore, slow temporal fluctuations in gastric-BOLD coupling are associated with changes in BOLD amplitude and occur simultaneously in all nodes. Thus, our results suggest that canonical RSNs based on instantaneous connectivity represent only one of the possible partitions of the brain into coherent networks based on temporal dynamics.

### Neural origin of gastric-BOLD coupling

SIr, SII and medial wall motor regions likely receive direct gastric inputs. The stimulation of the splanchnic (spinal) nerve that innervates the stomach evokes responses in contralateral SI and bilateral SII in several mammals (*Amassian, 1951*), and the spinothalamic tract was recently shown to target MWM in monkeys (*Dum et al., 2009*). Vagal stimulation can also evoke responses in somatomotor cortices (*Ito et al., 2003*). In addition, single neurons with convergent visceral and hand inputs have been observed in SI (*Brüggemann et al., 1997*), in line with the overlap we observed between the gastric network and hand representation. SIr, SII and MWM are not only targeted by documented ascending visceral pathways, they are also early nodes of the gastric network, with a phase

advance compared with that of other nodes. Thus, these areas could be the entry point of gastric afferences.

We found the right posterior insula, a region that receives direct cardiac inputs in monkeys (*Zhang et al., 1998*) and is considered as a visceral cortex, to be coupled with the stomach, with coupling similar to the weakest node of the gastric network. In addition, the right posterior insula appeared with a phase advance, in line with its role in visceroception. To be revealed, gastric-BOLD coupling in the right posterior insula required a region-of-interest approach, that is an increase of signal-to-noise ratio by averaging across voxels. The modest involvement of the insula in the present data might be due to the absence of an interoceptive task. Indeed, BOLD signal in the insula increases when participants explicitly monitor a visceral variable (see, e.g.[*Critchley et al., 2004*]).

Regions receiving direct visceral inputs are also early nodes of the gastric network. This suggests that the BOLD fluctuations locked to the gastric rhythm have a neural origin. An additional argument for a neural origin is that we found gastric-BOLD coupling in parieto-occipital regions, where neural activity in the alpha range is modulated by gastric phase (*Richter et al., 2017*). However, below, we examine the possibility that other non-neural mechanisms might contribute to gastric-BOLD coupling.

Artefactual BOLD fluctuations caused by head movements driven by stomach contractions seem unlikely. Indeed, gastric-BOLD coupling was neither related to head movement nor to EGG power that increases during stomach contractions. Another possibility is a vascular artifact. During digestion, gastric blood flow does indeed vary (*Matheson et al., 2000*), but cerebral blood flow is unaltered (*Gallavan et al., 1980*). Artificial distension of the stomach can cause increases in peripheral blood pressure (*Min et al., 2011*), but this peripheral increase is mostly due to the insertion of a bag catheter, not to its inflation (*Cantù et al., 2008*). Finally, spontaneous fluctuations in blood pressure in humans occur at approximately 0.1 Hz (so-called Mayer waves), which is much faster than gastric rhythm. Thus, a vascular effect seems unlikely, and the hypothesis that activity in the gastric network is driven by neural activity in areas directly receiving ascending inputs appears more plausible.

## What is the functional role of the gastric network?

Twenty years and thousands of articles after the discovery of the default network, the debate on its functional role at rest or during tasks is still open. Thus, any discussion of the functional role the gastric network can only be tentative and speculative at this stage. Several non-mutually exclusive interpretations can nevertheless be considered.

The functional role of the coupling between stomach and body maps might be related to homeostatic regulations, which would account for the partial overlap between the gastric network and regions involved in heart rate variability. More specifically, the gastric network might be involved in the regulation of digestion, which is accompanied by changes in cardiac output and heart rate (*Kelbaek et al., 1989*). In addition, the unusual experimental setting with abdominal electrodes and a moderate fasting state might have drawn participants' attention toward their internal state, notably of hunger. Since participants had been fasting for only 2 hr, their state of hunger was probably rather moderate and unlikely to have dominated their spontaneous thoughts for 20 min.

We find areas containing body maps in the gastric network. This could simply indicate that the stomach, as any other body part, such as for example the hand, is represented in any body map. The body maps of the gastric network are classically associated with different sensory modalities and resting-state networks. In addition to the primary and secondary somatosensory cortices, the gastric network includes MWM (CCZ, RCZp and SMA) that are involved in motor preparation and display a clear somatotopic organization (*Amiez and Petrides, 2014*; *Picard and Strick, 2001*). The gastric network also comprises the EBA, a functional region in the occipito-temporal cortex that selectively responds to visual images of the human body (*Downing et al., 2001*; *Weiner and Grill-Spector, 2011*) and is causally involved in body visual recognition (*Urgesi et al., 2007*), with a fine topographical organization (*Orlov et al., 2010*). The stomach, an organ that cannot be easily touched, moved or seen, thus appears to be mapped in body maps related to touching, moving or seeing the body. However, the areas where the stomach is represented are more multi-sensory than usually held. The EBA is not purely visual since it is also activated when participants move or imagine body parts without visual feedback (*Astafiev et al., 2004*), as well as during haptic recognition of body parts (*Kitada et al., 2009*; *Costantini et al., 2011*). Primary somatosensory cortex combines internal and external bodily information since it receives both tactile and visceral afferents

(*Brüggemann et al., 1997*; *Follett and Dirks, 1994*). Medial wall motor regions do not only contain motor maps but also receive visceral inputs (*Levinthal and Strick, 2012*). Thus, activity in 5 out of the 12 nodes of the gastric network could be simply explained by a representation of the stomach in the brain body maps.

However, the gastric network is not limited to body maps, it also comprises regions that play a role in mapping the external space in bodily coordinates, namely, the right superior parieto-occipital sulcus, dorsal precuneus and RSC. The superior parieto-occipital sulcus region is a visuo-motor area that encodes visual stimuli in bodily coordinates during action (*Bernier and Grafton, 2010*). The dorsal precuneus and RSC both implement the integration of information into an egocentric reference frame (i.e. centered on the body), a key basic mechanism involved in many different situations (*Burgess et al., 2001*; *Vann et al., 2009*), including foraging. All 12 nodes of the gastric network but three (the posterior cingulate sulcus and ventral and dorsal occipital clusters) either contain body maps or map external information in bodily coordinates. One could thus speculate that the gastric network coordinates these different body-centered maps. Indeed, the gastric rhythm is continuously produced and originates in the center of the body. In this view, the function of gastric-BOLD coupling in those nine areas would be to co-register body-centered maps of the body and of the external space.

In which type of tasks would the gastric network play a role? Foraging and feeding behaviors are likely candidates, since they involve both the coordination of different egocentric maps and the homeostatic regulation of digestion. Besides, in SI and EBA gastric-BOLD coupling is maximal in the hand and mouth region, suggesting a potential link with the stereotypical actions of feeding behavior, where food goes from hand to mouth, and from mouth to stomach. Still, the coordination of different systems of bodily coordinates is important for many actions besides feeding, such as navigating in the environment or grasping any object. Whether the gastric network plays a role in food-related, but also nonfood-related behaviors, remains to be determined.

## Delays and directionality of interactions

The gastric network is characterized by temporal fluctuations with delays between the gastric rhythm and brain regions. Delays in resting-state functional connectivity have been highlighted only recently (*Yellin et al., 2015*; *Mitra and Raichle, 2016*), but have long been documented in stimulus-induced BOLD responses (*Saad et al., 2001*; *Kruggel et al., 1999*). Within functionally coherent systems such as the visual (*Saad et al., 2001*) or auditory (*Kruggel et al., 1999*) systems, delays of 2 s are common. In this light, our finding of up to 3 s delays between areas much further apart thus not appear so surprising. Still, the interpretation of long delays is not straightforward. They are unlikely to directly reflect synaptic delays of fast sequential neural transmission between areas since feed-forward transfer, with only minimal local computations, can be as fast as 10 to 15 ms per processing stage (*Thorpe et al., 1996*). However, if local recurrent processing is involved, longer delays might occur. Delays might additionally reflect regional differences in the timing of the vascular response (*Saad et al., 2001*; *Kruggel et al., 1999*), in slow changes of neural activity over time, as in accumulation processes (*Yellin et al., 2015*), or in the involvement of neuromodulatory influences. The different factors may further be combined, that is neuromodulation might affect cerebrovascular reactivity (*Krimer et al., 1998*).

What is the directionality of the brain-stomach interactions? The methods used here do not allow to answer this question since PLV is not a directed measure. If the gastric network plays a role in homeostasis, interactions are likely to be bidirectional, because homeostasis implies both the monitoring of ascending inputs, to evaluate the peripheral state, and the production of descending control commands. Medial wall motor regions, which both receive inputs from the spino-thalamic tract, and generate sympathetic outputs, might fit with this schema. On the other hand, the gastric-locked modulation of the alpha rhythm in the ventral and dorsal occipital clusters was previously shown to be mostly due to ascending influences from the stomach to the brain (*Richter et al., 2017*).

## The gastric network is a novel resting-state network

RSNs have been defined as segregated systems that show synchronous fluctuations during rest (*Fox and Raichle, 2007*). The gastric network, albeit distinct from classical RSNs falls under this definition. In terms of dynamics, the gastric network is defined by its phase synchronization with the

stomach and its delayed connectivity between nodes. The gastric network can thus be considered a novel RSN that could not be previously observed due to methodological reasons.

As opposed to classical RSNs, the gastric network is characterized by delayed connectivity, with temporal delays that can extend to several seconds but are stable over time and captured by coherence and phase synchrony. Delays are an intrinsic characteristic of brain dynamics unfolding in anatomically connected networks (*Deco et al., 2011*) and pervasive even at the timescale of BOLD signal fluctuations (*Mitra et al., 2015*). Canonical RSNs based on instantaneous connectivity represent only one of the possible partitions of the brain into coherent networks based on temporal dynamics. Therefore, we propose the addition of delayed connectivity to the operational definition of RSNs.

## Materials and methods

### Experimental procedure

#### Participants

Thirty-four right-handed human participants took part in this study. All volunteers were interviewed by a physician to ensure the following inclusion criteria: absence of digestive, psychiatric or neurological disorders; BMI between 18 and 25; and compatibility with MRI recordings. Participants received a monetary reward and provided written informed consent for participation in the experiment and publication of group data. The study was approved by the ethics committee *Comité de Protection des Personnes Ile de France III* (approval identifier: 2007-A01125-48). All participants fasted for at least 90 min before the recordings. Data from four participants were excluded. Two were excluded because coughing artifacts caused excessive head movement during acquisition and corrupted the EGG data, and two were excluded because their EGG spectrum did not show a clear peak that could allow us to identify the frequency of their gastric rhythm. A total of 30 participants (mean age 24.2 ± SD 3.31, 15 females, mean BMI 21.48 ± SD 1.91) were included in the analysis described below. Because effect size was not known a priori, the study was powered to detect medium size effect (i.e. slightly above the median sample size of fMRI studies [*Poldrack et al., 2017*]).

#### MRI data acquisition

MRI was performed at 3 Tesla using a Siemens MAGNETOM Verio scanner (Siemens, Germany) with a 32-channel phased-array head coil. The resting-state scan lasted 900 s during which participants were instructed to lay still and fixate on a bull's eye on a gray background. A functional MRI time series of 450 volumes was acquired with an echo-planar imaging (EPI) sequence and the following acquisition parameters: TR = 2000 ms, TE = 24 ms, flip angle = 78°, FOV = 204 mm, and acquisition matrix = $68\times68 \times 40$ (voxel size = $3\times3 \times 3$ mm$^3$). Each volume comprised 40 contiguous axial slices covering the entire brain. High-resolution T1-weighted structural MRI scans of the brain were acquired for anatomic reference after the functional sequence using a 3D gradient-echo sequence (TE = 1.99 ms, TR = 5000 ms, TI-1 = 700 ms/TI-2=2500 ms, flip angle-1 = 4°/flip angle-2 = 5°, bandwidth = 240 Hz/pixel, acquisition matrix = $240 \times 256\times224$, and isometric voxel size = 1.0 mm$^3$). The anatomical sequence duration was 11 min 17 s. Cushions were used to minimize head motion during the scan.

#### Physiological signal acquisition

Physiological signals were simultaneously recorded during functional MRI acquisition using MRI compatible equipment. The electrogastrogram (EGG) and electrocardiogram (ECG) were acquired using bipolar electrodes connected to a BrainAmp amplifier (Brain products, Germany) placed between the legs of participants; the electrodes received a trigger signaling the beginning of each MRI volume. EGG was acquired at a sampling rate of 5000 Hz and a resolution of 0.5 μV/bit with a low-pass filter of 1000 Hz and no high-pass filter (DC recordings). ECG was acquired at a sampling rate of 5000 Hz and a resolution of 10 μV/bit with a low-pass filter of 1000 Hz and a high-pass filter of 0.016 Hz. Eye position and pupil diameter were recorded from the right eye with an EYELINK 1000 (SR Research, Canada) and simultaneously sent to BrainAmp amplifiers.

The skin of participants was rubbed and cleaned with alcohol to remove dead skin, and electrolyte gel was applied improve the signal-to-noise ratio. The EGG was recorded via four bipolar electrodes placed in three rows over the abdomen, with the negative derivation placed 4 cm to the left of the positive one. *Figure 1a* shows the electrode placement scheme. Electrodes covered a large portion of the left abdomen, to increase the chance of having an electrode close to the pacemaker of the stomach, located at the greater curvature of the mid to upper corpus (*O'Grady et al., 2010*). Because the gastric rhythm propagates from the pacemaker zone to the whole stomach, any electrode placed over the stomach (not necessarily over the pacemaker) will record the gastric rhythm, but with a possible delay. The midpoint between the xyphoid process and umbilicus was identified, and the first electrode pair was set 2 cm below this area, with the negative derivation set at the point below the rib cage closest to the left mid-clavicular line. Another electrode pair was set 2 cm above the umbilicus and aligned with the first electrode pair. The positive derivation of the third pair was set in the center of the square formed by electrode pairs one and two. The positive derivation of the fourth electrode pair was centered on the line traversing the xyphoid process and umbilicus at the same level as the third electrode. The ground electrode was placed below the lower left costal margin. The ECG was acquired using three bipolar electrodes that shared the same negative derivation, set at the third intercostal space. The positive derivations were set at the fifth intercostal space and separated by 4 cm.

Electrophysiological data were collected during fMRI data acquisition, as well as at least 30 s before and after. In addition, to rule out the possibility that the scanner pulse and B0 magnetic field could distort the frequency content of the EGG, a second EGG acquisition with an 8 min duration was performed after the acquisition of the MRI scans, with the participant positioned outside the tunnel of the scanner. Paired sample t-test was then performed to compare the peak frequencies obtained for each participant inside the scanner with those obtained outside the scanner for the same channels. This control analysis was run on 29 participants due to corrupted data in the EGG recordings outside the scanner tunnel in one participant.

## MRI preprocessing

Brain imaging data were preprocessed using Matlab (Matlab 2013b, MathWorks, Inc., United States) and the Statistical Parametric Mapping toolbox (SPM 8, Wellcome Department of Imaging Neuroscience, University College London, U.K.). Images of each individual participant were corrected for slice timing and motion with six movement parameters (three rotations and three translations). Two participants who moved more than 3 mm during the functional scan were excluded from the study. Each participant's structural image was normalized to Montreal Neurological Institute (MNI) space of 152 participants' average T1 template provided by SPM with affine registration followed by nonlinear transformation (*Ashburner and Friston, 1999*; *Friston et al., 1995*). The normalization parameters determined for the structural volume were then applied to the corresponding functional images. The functional volumes were spatially smoothed with a 3 mm$^3$ full-width half-maximum (FWHM) Gaussian kernel. The time series of voxels inside the brain, as determined using an SPM a priori mask, were subjected to the following preprocessing steps using the FieldTrip toolbox (*Oostenveld et al., 2011*) (Donders Institute for Brain, Cognition and Behaviour, Radboud University Nijmegen, the Netherlands. See http://www.ru.nl/neuroimaging/fieldtrip, release 01/09/2014). Linear and quadratic trends from each voxel's time series were removed by fitting and regressing basis functions, and we bandpass filtered the BOLD time series between 0.01 and 0.1 Hz using a fourth order Butterworth infinite impulse response filter. A correction for cerebrospinal fluid motion was obtained by regressing out the time series of a 9 mm diameter sphere located in the fourth ventricle (MNI coordinates of the center of the sphere [0 –46 −32]).

## EGG preprocessing

Data analysis was performed using the FieldTrip toolbox. Data were low-pass filtered below 5 Hz to avoid aliasing and downsampled from 5000 Hz to 10 Hz. To identify the EGG peak frequency (0.033–0.066 Hz) for each participant, we computed the spectral density estimate at each EGG channel over the 900 s of an EGG signal acquired during the fMRI scan using Welch's method on 200 s time windows with 150 s overlap. Spectral peak identification was based on the following criteria: peaking power larger than 15μV$^2$ and sharpness of the peak. Two participants were excluded from

further analysis at this stage because their spectral peak was not well defined, with a power smaller than 15μV$^2$. In 20 participants, peaking power was the largest at the EGG electrode with the best defined spectral peak. In 10 participants, we used the second most powerful channel because the spectral peak was sharper. Data from the selected EGG channel were then bandpass filtered to isolate the signal related to gastric basal rhythm (linear phase finite impulse response filter, FIR, designed with Matlab function FIR2, centered at EGG peaking frequency, filter width ±0.015 Hz, filter order of 5). Data were filtered in the forward and backward directions to avoid phase distortions and downsampled to the sampling rate of the BOLD acquisition (0.5 Hz). Filtered data included 30 s before and after the beginning and end of MRI data acquisition to minimize ringing effects.

MR gradient artifacts affect the electrophysiological signal down to approximately 10 Hz, which is far above EGG frequency (~0.05 Hz). Thus, no specific artifact gradient procedure was necessary. We further checked that EGG frequency inside and outside the scanner did not differ (see Results).

## Data analysis

### Quantification of gastric-BOLD phase synchrony

The BOLD signals of all brain voxels were bandpass filtered with the same filter parameters as the ones used for the EGG preprocessing. The first and last 15 volumes (30 s) were discarded from both the BOLD and EGG time series. The updated duration of the fMRI and EGG signals in which the rest of the analysis was performed was 840 s. The Hilbert transform was applied to the BOLD and EGG time series to derive the instantaneous phases of the signals. The PLV (*Lachaux et al., 1999*) was computed as the absolute value of the time average difference in the angle between the phases of the EGG and each voxel across time (*Equation 1*).

$$PLV_{x,y} = \left| \frac{1}{T} \sum_{t=1}^{T} e^{i(\varnothing x(t) - \varnothing y(t))} \right| \qquad (1)$$

where T is the number of time samples, and x and y are the two time series. The PLV measures phase synchrony irrespective of temporal delays and amplitude fluctuations and is bounded between 0 (no synchrony) and 1 (perfect synchrony). Two pure sinewaves at the same frequency will thus always have a PLV of 1. However, the stomach is not perfectly regular, and the EGG is not a perfect sinewave. The phase-locking procedure identifies BOLD regions that go faster when the stomach goes faster, and slower when the stomach goes slower. The bandpass filter we use is large enough to retrieve all those fluctuations. The PLV was first assessed over the whole duration of the recording. In a second step, we computed the time-varying PLV in a 60 s time window shifted by 10 s.

### Statistical procedure for determining regions showing significant gastric-BOLD coupling at the group level

We employed a two-step statistical procedure adapted from a previous work (*Richter et al., 2017*). We estimated chance-level gastric-BOLD coupling at each voxel and in each participant. We then used group-level statistics to determine regions in which gastric-BOLD coupling was greater than chance.

We first estimated the chance-level PLV at each voxel for each participant. We created surrogate datasets in which the phase relationship between the EGG and BOLD time series was disrupted by offsetting the EGG time series with respect to the BOLD time series. In other word, acceleration/deceleration in the EGG are no longer aligned with the acceleration/deceleration in the BOLD. In practice, the EGG time series was shifted by a time interval of at least ±60 s (i.e. approximately 3 cycles of the gastric rhythm) with respect to the BOLD time series. Data at the end of the recording were wrapped to the beginning. Given the 420 samples in the BOLD time series, this procedure generated 360 surrogate datasets from which we could compute the distribution of the chance-level PLV for each voxel in each participant. The chance-level PLV was defined as the median value of the chance-level PLV distribution for each voxel and participant. Because the amplitudes of the series in the surrogate datasets are identical to the original ones, any bias due to signal amplitude is present in both original and surrogate datasets. Note that PLV is a measure that depends on sample size (*Vinck et al., 2010*; *Bastos and Schoffelen, 2015*). Here, we can safely compare PLV values between the original and surrogate datasets in each participant because the original and surrogate

datasets contain exactly the same number of samples (420 samples in all datasets in all participants). We defined coupling strength as the difference between the empirical PLV and chance-level PLV.

In a second step, we tested whether the empirical PLV differed from the chance-level PLV across participants. We used a cluster-based permutation procedure (*Maris and Oostenveld, 2007*), as implemented in FieldTrip (*Oostenveld et al., 2011*), that extracts clusters of voxels showing significant differences at the group level while intrinsically correcting for multiple comparisons. This non-parametric method is exempt from the high rate of false positives associated with the Gaussian shape assumption often present in fMRI studies (*Eklund et al., 2016*). The procedure consists of comparisons between the empirical PLV and chance-level PLV across participants using t-tests at each voxel. Candidate clusters are formed by neighboring voxels exceeding the first-level t-threshold (p<0.01, two-sided). Each candidate cluster is characterized by the sum of the t-values in the voxels defining the cluster. To determine the sum of t-values that could obtained by chance, we computed a cluster statistics distribution under the null hypothesis by randomly shuffling the labels 'empirical' and 'chance level' 10,000 times and applied the clustering procedure. At each permutation, we retained the largest positive and smallest negative summary statistics obtained by chance across all voxels and thus built the distribution of cluster statistics under the null hypothesis and assessed the empirical clusters for significance. Because the maximal values across the whole brain are retained to build the distribution under the null hypothesis, this method intrinsically corrects for multiple comparisons. Clusters are characterized by their summary statistics (sum(abs(t))) and Monte-Carlo p value. Clusters with a Monte-Carlo p value<0.05 (two-sided, corrected for multiple comparisons) were considered significant and are reported in the Results section as nodes of the gastric network.

As an additional control, we computed gastric-BOLD coupling at each voxel between the BOLD data of the participant and the EGG data of the other 29 participants. Chance-level PLV was defined as the median of the 29 surrogate PLVs, and compared to empirical PLV using the clustering method described above. Note that in this case, chance level PLV is estimated from only 29 surrogate data sets, as compared to 360 surrogate data sets in time-shift approach, resulting in a less precise estimate.

## Quantification of gastric-bold shared variance

To estimate the amount of variance in the BOLD signal that could be accounted for by gastric coupling, we computed the squared coherence coefficient between the EGG and average BOLD time course across all voxels in each significant cluster using FieldTrip software. The coherence coefficient measures phase and amplitude consistency across time and is a frequency domain analog of the cross-correlation coefficient in the temporal domain. Therefore, its squared value can be interpreted as the amount of shared variance between two signals at a certain frequency (*Bastos and Schoffelen, 2015*). First, we estimated the frequency spectrum of the full-band (0.01–0.1 Hz) EGG and BOLD signals (Welch method on a 120 s time window with 20 s overlap). We then computed the coherence coefficient between the spectrum of each participant's EGG and each cluster's time series at gastric frequency ($\omega$) as the absolute value of the product of the amplitudes (A) of the signals and their phase (φ) difference averaged across time windows (t) and normalized by the square root of the product of their squared amplitudes averaged across time windows.

$$coh_{xy}(\omega) = \frac{\left|\frac{1}{T}\sum_{t=1}^{T} A_x(w,t)A_y(w,t)e^{i(\varphi x(w,t)-\varphi y(w,t))}\right|}{\sqrt{\left(\frac{1}{T}\sum_{t=1}^{T} A_x^2(w,t)\right)\left(\frac{1}{T}\sum_{t=1}^{T} A_y^2(w,t)\right)}} \qquad (2)$$

The coherence coefficient was then squared and averaged across participants such that the final group value represented the shared variance between the EGG and each cluster BOLD activity at the normogastric peak.

## Between-participant phase-delay consistency

To quantify temporal delays in the gastric network, we ran group-level analysis on the gastric-BOLD phase-locking angle. In each participant, we first computed a mean BOLD time series per node by averaging the voxel time series in each significant cluster. We then computed the relative phase-

locking angle $\phi_{k\ relative}$ of the node $k$ between the node time series $x$ and the EGG $y$ using *equation 3*, where $\phi_{k\ relative}$ corresponds to the phase-locking angle $\phi_k$ of node k with respect to the EGG minus the average angle across all nodes. $\phi_{k\ relative}$ thus quantifies the phase advance or lag of each node relative to the gastric network. We analyze relative, rather than absolute phase values, because of there might be a constant but unknown phase delay between the recorded EGG and the rhythm of the gastric pacemaker.

$$\Phi_k = \arg\left(\frac{1}{T}\sum_{t=0}^{T} e^{i(\phi x(t) - \phi y(t))}\right) \tag{3}$$

$$\Phi_k\ relative = \Phi_k - \frac{1}{K}\sum_{k}^{K}\Phi_k \tag{4}$$

Between-participant phase-delay consistency was then obtained at each node by averaging the unit vectors of the relative phase-locking angles across $P$ participants using Equation 5.

$$Between-participant\,phase-delay\,consistency_k = \left|\frac{1}{P}\sum_{P=1}^{P} e^{i(\phi k)}\right| \tag{5}$$

To determine whether there were significant differences across the angle of gastric network clusters, we submitted the values of each node and participant's relative phase-locking angle to Watson-Williams test, a circular analog of one-way ANOVA for circular data, using the circstat Matlab toolbox (*Berens, 2009*)

## Functional connectivity: correlation and coherence

FC was defined as shared variance and computed using either the squared Pearson correlation coefficient or squared coherence. We computed the Pearson correlation between the bandpass-filtered BOLD time series (gastric peaking frequency ± 0.015 Hz) averaged across voxels in each gastric node, as well as in two control regions outside the gastric network, the right ventral precuneus and right ventral insula. The ventral precuneus, a core node of the default network, was defined using a 9 mm$^3$ ROI centered in the coordinates provided by *Fox et al. (2005)* (MNI x=-5 y = −52.5 z = 41). The right ventral insula ROI was provided by the parcellation performed by *Deen et al. (2011)*.

To compute coherence between BOLD time series, we first estimated the frequency spectrum of the full-band (0.01–0.1 Hz) BOLD time series using the Welch method with 36 time windows of 120 s with 20 s overlap. We then computed coherence using the FieldTrip implementation of equation number two and used the squared coherence at the gastric peak frequency of each participant as an estimate of shared variance.

## Heart rate variability analysis

We first removed the MRI gradient artefact from the ECG data using the FMRIB plug-in (*Iannetti et al., 2005*; *Niazy et al., 2005*, version 1.21) for EEGLAB (*Delorme and Makeig, 2004*, version 14.1.1), provided by the University of Oxford Centre for Functional MRI of the Brain (FMRIB). Data from the three ECG channels was then 1–100 Hz bandpass filtered using a FIR filter, designed with Matlab function firws. We then retrieved the inter-beat-interval (IBI) time series by identifying R peaks using a custom semi-automatic algorithm, which combined automatic template matching with manual selection of R peaks for extreme IBIs. This procedure was performed in the ECG channel of each participant that required the least manual identification of R peaks. The resulting IBI time series were then interpolated at 1 Hz using a spline function (order 3), and band-pass filtered at high (0.04–0.15 Hz) and low frequencies (0.15–0.4 Hz) of heart rate variability using a FIR filter (designed with Matlab function FIR2, center frequency for LFHRV 0.1 Hz ± 0.06 Hz, HFHRV centered at 0.275 ± 0.125 Hz) and then downsampled at MRI frequency (0.5 Hz). The amplitude envelope of HF- and LF-HRV were then computed using the Hilbert transform and used as regressors of interest (without convolution with the HRF as in [*Critchley et al., 2003*]) in two separate first level GLMs, which also included six movements parameters as regressors. The MRI pre-processing parameters were the same as for the gastric-BOLD coupling analysis (slice-timing and motion correction, co-

registration to MNI space and spatial smoothing of FWHM = 3 mm). The BOLD time series were high-pass filtered (cutoff: 128 s) for the GLM analysis. GLM analysis was performed using SPM8 (*Friston et al., 1994*).

Contrast images from the first level were entered into two separate second level random-effects analysis to test for consistent effects across the 30 participants separately for HF and LF HRV. The contrast images were spatially smoothed (FWHM = 8 mm), and submitted to a one-sample T-test. Statistical inference was performed at the voxel level, family-wise-error-corrected ($p_{FWE} < 0.05$) for multiple comparisons over the whole brain.

### Pupil diameter analysis

Pupil size during blinks and saccades (as automatically detected by the EyeLink software) was estimated by interpolating between pupil size 100 ms before and 100 ms after each event. Artefacted windows separated by less than 200 ms were combined and treated as a single epoch. Data from seven participants were excluded due to a high (>20%) amount of artefacted data. Data from three participants were excluded because MRI and pupil data could not be synchronized, due to missing triggers. Pupil data from the remaining 20 participants were downsampled at MRI frequency (0.5 Hz), bandpassed filtered (0.0078–0.1 Hz) using a butterworth infinite impulse response filter and used as a regressor (convolved with the canonical HRF as in [*Yellin et al., 2015*; *Schneider et al., 2016*]) in a first level GLM, which also included six movement nuissance regressors. The BOLD time series were high-pass filtered (cutoff: 128 s) for the GLM analysis (SPM8). Contrast images from the first level entered into a second level random-effects analysis to test for consistent effects of pupil size across the 20 participants. The contrast images were spatially smoothed (FWHM = 8 mm), and submitted to a one-sample T-test. Statistical inference was performed at the voxel level, (p<0.001, uncorrected for multiple comparisons).

### Bayes factor

Bayesian statistics on correlation coefficients were computed and interpreted according to (*Wetzels and Wagenmakers, 2012*) and (*Kass and Raftery, 1995*), and Bayesian statistics on two sample (unpaired) comparisons according to (*Rouder and Morey, 2011*). Regarding the specific test of an absence of effect of voxel motion susceptibility on coupling strength (H0), submillimeter voxel motion was estimated as in (*Power et al., 2012*), and H1 was modeled as the minimum effect size required to detect a significant difference from zero, given one-sample t-test of 29 degrees of freedom on a normal distribution with a mean of 0 and a standard deviation of 1. The same method was used to test for the absence of a difference between the EGG peak frequency measured inside and outside the scanner.

### Anatomical and functional overlays and meta-analysis

Functional group-level images were overlaid on a 3D rendering of the MNI template using MRIcroGL (https://www.nitrc.org/frs/?group_id=889, June 2015). The results from the literature were converted when necessary from Talairach coordinates to MNI coordinates using the nonlinear transform provided by (http://imaging.mrc-cbu.cam.ac.uk/imaging/MniTalairach) and visualized using Caret software (*Van Essen et al., 2001*) (http://www.nitrc.org/projects/caret/, v5.65). Overlap of the gastric network with cytoarchitectonic subdivisions of primary and secondary somatosensory cortices was determined with the anatomy toolbox for SPM (*Eickhoff et al., 2005*; *Eickhoff et al., 2007*; *Eickhoff et al., 2006*).

### Nifti overlays availability

Unthresholded t maps of empirical vs chance PLV comparisons (intermediate step for *Figure 2a*), mask of significant clusters (*Figure 2a*), unthresholded and significant mask of HF and LF HRV and pupil diameter (*Figure 3*) and average phase-locking angle of each significant cluster (*Figure 4b*) are available at Neurovault (*Gorgolewski et al., 2015*) at the following address: http://neurovault.org/collections/GMHHGEXA/

## Custom code

The custom code used for this article (*Rebollo, 2018*) can be accessed online at the following address: https://github.com/irebollo/stomach_brain_Scripts. A copy is archived at https://github.com/elifesciences-publications/stomach_brain_Scripts.

## Acknowledgements

This work was supported by the European Research Council (ERC) under the European Union's Horizon 2020 research and innovation program (grant agreement No 670325) to CT-B, as well as by ANR-10-LABX-0087 IEC and ANR-10-IDEX-0001–02 PSL*. IR was supported by a grant from DIM Cerveau et Pensée and Fondation Bettencourt-Schueller. We thank Margaux Romand-Monnier for help during data acquisition.

## Additional information

### Funding

| Funder | Grant reference number | Author |
| --- | --- | --- |
| H2020 European Research Council | 670325 | Catherine Tallon-Baudry |
| Agence Nationale de la Recherche | ANR-10-LABX-0087 IEC | Catherine Tallon-Baudry |
| DIM cerveau et pensee | | Ignacio Rebollo |
| Fondation Bettencourt Schueller | | Ignacio Rebollo |
| Agence Nationale de la Recherche | ANR-10-IDEX-0001-02 PSL* | Catherine Tallon-Baudry |

The funders had no role in study design, data collection and interpretation, or the decision to submit the work for publication.

### Author contributions

Ignacio Rebollo, Conceptualization, Data curation, Formal analysis, Investigation, Visualization, Methodology, Writing—original draft, Writing—review and editing; Anne-Dominique Devauchelle, Data curation, Software, Investigation, Made substantial contributions to the acquisition and analysis of the data, Revised the article and provided approval for the final version of the article; Benoît Béranger, Investigation, Made substantial contributions to the acquisition of the data, Revised the article and provided approval for the final version of the article; Catherine Tallon-Baudry, Conceptualization, Data curation, Supervision, Funding acquisition, Methodology, Writing—original draft, Project administration, Writing—review and editing

### Author ORCIDs

Ignacio Rebollo http://orcid.org/0000-0002-4119-9955
Catherine Tallon-Baudry http://orcid.org/0000-0001-8480-5831

### Ethics

Human subjects: Participants received provided written informed consent for participation in the experiment. The study was approved by the ethics committee Comité de Protection des Personnes Ile de France III

### Decision letter and Author response

Decision letter https://doi.org/10.7554/eLife.33321.022
Author response https://doi.org/10.7554/eLife.33321.023

## Additional files

**Supplementary files**
• Transparent reporting form
DOI: https://doi.org/10.7554/eLife.33321.020

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
