## [Decision Letter]

Thank you for submitting your article "Stomach-brain synchrony binds neural representations of the body in a novel, delayed-connectivity resting-state network" for consideration by *eLife*. Your article has been reviewed by three peer reviewers, and the evaluation has been overseen by a Reviewing Editor and David Van Essen as the Senior Editor. The following individual involved in review of your submission has agreed to reveal his identity: Hugo Critchley (Reviewer #1).

The reviewers have discussed the reviews with one another and the Reviewing Editor has drafted this decision to help you prepare a revised submission.

Summary:

Rebollo and colleagues measured phase coupling between the blood-oxygen level dependent (BOLD) signal and stomach slow-wave potentials in resting humans. In their manuscript, the authors describe a novel, elegant experiment through which they achieved this goal obtaining an interesting set of findings. Simultaneous functional magnetic resonance imaging (fMRI) and electrogastrography (EGG) recordings revealed that the time series of stomach slow waves and the time series of BOLD signal are significantly phase-coupled in twelve different voxel clusters scattered across the brain – a set of clusters that the authors call "gastric network". In these clusters the empirically measured phase-locking value (PLV) of the two-time series is significantly different from the median value of simulated PLVs that could be expected by chance.

The empirical PLV is computed plugging the amplitude-normalised, Fourier-transformed values of the two series into the classical equation for coherence. In turn, squared coherence is used to estimate of shared variance between the BOLD signal and slow-wave potentials: on average, between 12% and 16.9%. The authors used PLV to calculate the relative phase-locking angle of each node of the gastric network – a measure of the lag between an event in the slow-wave cycle and the corresponding change of BOLD phase in a voxel cluster. This led them to conclude that some gastric network nodes are up to 3.3 s slower to adjust their phase to slow-wave potentials than other nodes. Such delays are accompanied by corresponding delays in between-node functional connectivity. Thus, in contrast with classical resting-state networks, the gastric network nodes are related to each other not only through instantaneous flows of current, but also through delayed connectivity.

The reviewers were impressed at the technical achievements described and agreed on the good aspects, potential interest and significance of this MS. The reviewers also recognised the need for some additional informative analyses. BTW I did not comment on the lack of Insula activity because that was the first and most striking observation!

Essential revisions:

1) Testing phase timing relationships that take into account any temporal differences seems counterintuitive: Band-pass filtering at a specific frequency (here around 3 cycles per min) would reveal areas showing that frequency: The difference between high and low PVL reflects the consistency with which a region expresses that frequency, rather than necessarily any implied causal relationship perhaps. Can the authors present a clearer picture concerning causal relationships between EGG and brain haemodynamics?

2) In establishing the individual specificity of the relationships, the authors chose not to randomise across participants the testing of EGG (from other participants) PSVs with brain haemodynamics: Would this be a more or less powerful demonstration of the individual specificity of brain coupling to gastric rhythm?

3) The authors make a case for the gastric coupling to be unrelated to typical autonomic regulation areas, including insula, based on citation of a meta-analysis of largely task evoked changes in cardiovascular and or electrodermal autonomic responses. While this meta-analysis does include some HRV data and resting studies, they are predominantly focused on sympathetic cardiovascular arousal, which may be associated with EGG suppression. The author's acquired heart rate and pupil data on their participants, suggesting they could either test within their data for autonomic overlap with EEG-coupled areas on an individual basis, or test if EEG frequency rhythms apparent within these autonomic data series.

4) The measurement of slow fluctuations within already slow haemodynamic signal changes and suggesting that seconds-long lags between regions that argued to be functionally connected are both somewhat removed from the heuristic model of <10ms-a-synapse processing and representation of sensory information within and between brain regions. It is for example possible that the neural input signal is modulating cerebrovascular reactivity /tonicity through intrinsic neuromodulatory systems rather than reflecting neuron to neuron information exchange. If the gastric input anchors body centered representations within a single functional system, the differential regional expression of significant time lags requires more mechanistic explanation.

5) Related to this is the controversial claim is the statement that the newly discovered coupling between gastric rhythm and resting-state BOLD signal "appears to bind together distributed maps encoded in bodily coordinates into a coherent functional system (subsection “Gastric network and bodily space”)".

This assertion should be discussed in more detail. First, the very idea that synchronous oscillations serve a high-level cognitive purpose as binding may not be completely proved, as the unresolved debate over binding-by-γ-synchrony shows (Merker, 2013; Ray and Maunsell, 2015). Rather than acting as an information-processing neural code of a sort, γ oscillations may simply be a neural signature of the balance between excitation and inhibition. Likewise, the EGG-BOLD synchrony found by the authors may reflect general homeostatic processes which do not play a particular role in cognition. Cerebral blood flow supplies nutrients to neurons, and such nutrients come from digestive processes. In this context, the slow-wave potentials of interstitial cells of Cajal may simply trigger food intake, given the fact that participants fasted for at least 90 minutes before the experiment (Subsection “Experimental procedure”). This could explain why the gastric network so neatly overlaps with areas mapping faces and hands in general (OP1 and OP4: subsection “The gastric network includes body maps associated with touch, action and vision”), the mouth (EBA: subsection “The gastric network includes body maps associated with touch, action and vision”), teeth, lips, tongue, and hand fingers (right SI: subsection “The gastric network includes body maps associated with touch, action and vision”), as well as with three medial wall motor preparation regions (CCZ, RCZp, and SMA: subsection “The gastric network includes body maps associated with touch, action and vision”) which become more active when subjects are asked to move their tongue or their hand (Amiez and Petrides, 2014). The authors previously excluded that variations in cerebral blood flow could be due to digestion, although they backed their position with a single 1980 study on regional blood flow in the conscious dog (Gallavan et al., 1980). Nevertheless, they should consider the opposite hypothesis, namely, that the lack of nutrients to digest may spawn a long-range synchrony between the stomach and brain areas related to food intake. Data on caloric intake prior to the experiment, blood ghrelin levels, or a self-reported hunger scale may have shed light on the issue. Thus, I recommend the authors discuss and rule out more mundane interpretations before asserting that "stomach-brain synchrony binds neural representations of the body".

6) The interesting, newly-discovered synchrony arises in a resting state: In a task-free environment, what is the functional advantage of having "different body-centered representations" anchored by a delayed-connectivity gastric network (subsection “The gastric network is a novel resting-state network”) with lags up to 3.3 seconds? Of note, previous "binding-by-synchrony" proposals were based on task-dependent results.

7) Only 5 out of 12 nodes of the gastric network have a clear somatotopic organisation (subsection “Experimental procedure”): thus, "binding" interpretation of the authors does not explain the role that the majority of the nodes play in the network. Even the nodes which do have a somatotopic arrangement contain neurons tuned to the external perception of body parts, be it touch (nodes partially overlapping with SI and SII), motor preparation (MWM) or vision (EBA). As the authors concede, the stomach is "an organ that cannot be easily touched, moved, or seen (subsection “Gastric network and bodily space”)". It is intriguing and difficult to understand why the network revealed here seems organized around brain areas involved in body and bodily space as well as the stronger claim that it is gastric rhythm by itself that functionally links these areas.

8) The authors make the point that SI, SII and MWM "likely" have direct gastric input, hence their claim that gastric-BOLD coupling has a neural origin (subsection “Neural origin of gastric-BOLD coupling”). Yet the insular cortex is almost completely missing from the gastric network (subsection “Marginal gastric-brain coupling in the insula and autonomic control networks”). This is surprising, given that the insula is certainly linked to the stomach and likely supports representations of the whole body (Craig, 2002, 2009). The authors should advance a clear hypothesis to explain this seemingly counter-intuitive finding which relates to the real meaning of the brain-stomach synchronization.

9) The authors should discuss the possibility that unusual measuring circumstances, wherein the participants lie in the scanner with electrodes fitted on their stomach area, create unusual conditions leading heightened awareness of the own body. This is not to say that activity in all nodes of the current network is directly and primarily related to body awareness. Indeed, some areas (SI, SII) receive direct gastric input. The network presented here may be less homogeneous than argued here and that activity in some of the nodes may have multiple bases. This would particularly seem to be the case for EBA and one can imagine different explanations for activity in EBA and for the fact that it comes on the latest. In the same vein, some arguments in defense of the representational specificity of the network appear overly speculative, e.g. suggestions about the spatial aspects of the body representation function.

---

## [Author Response]

Summary:Rebollo and colleagues measured phase coupling between the blood-oxygen level dependent (BOLD) signal and stomach slow-wave potentials in resting humans. In their manuscript, the authors describe a novel, elegant experiment through which they achieved this goal obtaining an interesting set of findings. Simultaneous functional magnetic resonance imaging (fMRI) and electrogastrography (EGG) recordings revealed that the time series of stomach slow waves and the time series of BOLD signal are significantly phase-coupled in twelve different voxel clusters scattered across the brain – a set of clusters that the authors call "gastric network". In these clusters the empirically measured phase-locking value (PLV) of the two-time series is significantly different from the median value of simulated PLVs that could be expected by chance.The empirical PLV is computed plugging the amplitude-normalised, Fourier-transformed values of the two series into the classical equation for coherence. In turn, squared coherence is used to estimate of shared variance between the BOLD signal and slow-wave potentials: on average, between 12% and 16.9%. The authors used PLV to calculate the relative phase-locking angle of each node of the gastric network – a measure of the lag between an event in the slow-wave cycle and the corresponding change of BOLD phase in a voxel cluster. This led them to conclude that some gastric network nodes are up to 3.3 s slower to adjust their phase to slow-wave potentials than other nodes. Such delays are accompanied by corresponding delays in between-node functional connectivity. Thus, in contrast with classical resting-state networks, the gastric network nodes are related to each other not only through instantaneous flows of current, but also through delayed connectivity.The reviewers were impressed at the technical achievements described and agreed on the good aspects, potential interest and significance of this MS. The reviewers also recognised the need for some additional informative analyses. BTW I did not comment on the lack of Insula activity because that was the first and most striking observation!

We thank the reviewers for their time and positive comments, as well as for their useful suggestions to improve data analysis and discussion. Briefly, we performed all required controls and additional data analysis, notably a refined analysis of the insula and analysis of HRV and pupil diameter. We have also rewritten large sections of the Discussion section, in particular to offer a more balanced and comprehensive view of the different possible functional roles of the gastric network (trying to synthetize all comments from the reviewers on this issue, i.e., points 5, 6,7,9), and to discuss the possible origins of the long delays observed here.

We respond to each point in detail below.

Essential revisions:1) Testing phase timing relationships that take into account any temporal differences seems counterintuitive: Band-pass filtering at a specific frequency (here around 3 cycles per min) would reveal areas showing that frequency: The difference between high and low PVL reflects the consistency with which a region expresses that frequency, rather than necessarily any implied causal relationship perhaps. Can the authors present a clearer picture concerning causal relationships between EGG and brain haemodynamics?

We are measuring temporal (phase) differences that are consistent over gastric cycles, without any a priori on the value of the temporal difference. One of the reasons why we chose this approach is that the phase of the gastric rhythm recorded by external, cutaneous electrodes is arbitrary. This important information was missing in the previous version. Briefly, the gastric rhythm is mostly generated in the pacemaker zone located at the greater curvature of the mid to upper corpus and propagates to the rest of the stomach with delays (O'Grady et al., 2010). Depending on where the electrodes are located with respect to the pacemaker zone, we might pick the gastric rhythm with a delay specific to a given participant and electrode. Hence, the measure of interest is the consistency of the delay between BOLD and EGG across gastric cycles, rather than an absolute delay value. This is now specified in subsection “EGG preprocessing”). Note that this is also the reason why we analyze relative, rather than absolute, delays between nodes (now explained in subsection “Between-participants phase-delay consistency”).

The second issue raised in the comment is that "band-pass filtering would reveal areas showing gastric frequency". The band passed EGG and BOLD data contain information about amplitude and phase. PLV is sensitive to phase, but not amplitude (as now specified when introducing PLV in the Results section). We nevertheless verified that coupling strength and BOLD power at gastric frequency were not correlated (Bayes factor<0.001, indicating decisive evidence for the absence of a link between coupling strength across the brain and BOLD power at gastric frequency). This result has been added to subsection “Partial overlap between gastric network and autonomic networks”.

In addition, while we used a procedure validated in a previous study (Richter et al., 2017), we realized that the procedure was not fully explained in the previous version of this article. We have modified the Material and methods section to be more explicit. Briefly, the whole method relies on the fact that the stomach is not perfectly regular, and the EGG is not a perfect sine wave. The phase-locking procedure identifies BOLD regions that go faster when the stomach goes faster, and slower when the stomach goes slower. The band pass filter we use is large enough to retrieve all those fluctuations. In the surrogate dataset, the phase link between the BOLD and the EGG is broken because the two-time series are no longer temporally aligned. Furthermore, any bias due to power of the BOLD at gastric frequency is also present in the surrogate datasets and is thus taken into account.

Last, regarding causality: PLV reveals consistent temporal relationships between two-time series, independently from amplitude, but does not say anything about the directionality of interactions. This is now specified when introducing PLV in the Results section. We have also added a full paragraph in the Discussion section about the directionality of interactions.

2) In establishing the individual specificity of the relationships, the authors chose not to randomise across participants the testing of EGG (from other participants) PSVs with brain haemodynamics: Would this be a more or less powerful demonstration of the individual specificity of brain coupling to gastric rhythm?

We opted for randomizing within participants for two reasons: first, this is a more conservative approach, since the only difference between empirical and surrogate datasets is BOLD-EGG phase relationships. Second, we can create a large number of surrogate datasets (360), to obtain a good estimate of chance level PLV.

Still, we ran this control and now also report the gastric-network when estimating chance-level PLV by computing gastric-BOLD coupling between the BOLD signal of one participant with the EGG of the other 29 participants. Although this approach should be more liberal (i.e., EGG frequency and power in the surrogate data are different from the original data) and less powerful (we estimate chance level PLV in each participant from only 29 surrogate data sets), the results are qualitatively similar, with coupling occurring either in the same or neighboring voxels. This additional control is presented in Figure 2, and the Material and methods section updated.

3) The authors make a case for the gastric coupling to be unrelated to typical autonomic regulation areas, including insula, based on citation of a meta-analysis of largely task evoked changes in cardiovascular and or electrodermal autonomic responses. While this meta-analysis does include some HRV data and resting studies, they are predominantly focused on sympathetic cardiovascular arousal, which may be associated with EGG suppression. The author's acquired heart rate and pupil data on their participants, suggesting they could either test within their data for autonomic overlap with EEG-coupled areas on an individual basis, or test if EEG frequency rhythms apparent within these autonomic data series.

- Heart-rate variability. We identified the brain regions associated with high and low frequency heart rate variability in our data, at rest (subsection “Partial overlap between gastric network and autonomic networks”). Briefly, the overlap between the gastric network with HRV regions is larger (30%) than previously observed with metanalytic data. The overlap occurs mostly in medial motor regions and in the posterior cingulate cluster, and, to a lesser extent, in the dorsal occipital clusters. We now present those results in a new Figure 3A. The overlap between a pure sympathetic measure (EDA) from the meta-analysis of Beissner et al., is presented in Figure 3—figure supplement 1, as well as the comparison between the resting state HRV (HF and LF) and meta-analytic HF-HRV and EDA. Note that in this new analysis we do find a link between low-frequency HRV and insula, as expected from the literature. We have modified the discussion to acknowledge the partial overlap between the gastric network and heart-rate variability regions.

- Pupil. As now reported in the Results section:

" We also determined brain regions that correlate with pupil diameter (n=20 due to data loss or artefacts; Figure 3B). The strongest correlations were found in occipital regions, somato-motor cortices and medial wall motor regions. 17% of the gastric network (SI, SIIr, MWM and EBA) overlaps with regions correlating with pupil diameter. Shared variance between pupil diameter and EGG, estimated from squared coherence, was 9.7% ± 2.5%. Coupling strength averaged across SI, SIIr, MWM and EBA did not correlate with shared pupil-EGG variance (mean r= 0.05, p=0.82, BF= 0.17 which indicates substantial evidence for the null hypothesis). "

These two new analysis led us to reconsider the potential link between gastric-BOLD coupling and homeostatic regulations, as reflected in the new section of the discussion entitled "What is the functional role of the gastric network?".

4) The measurement of slow fluctuations within already slow haemodynamic signal changes and suggesting that seconds-long lags between regions that argued to be functionally connected are both somewhat removed from the heuristic model of <10ms-a-synapse processing and representation of sensory information within and between brain regions. It is for example possible that the neural input signal is modulating cerebrovascular reactivity /tonicity through intrinsic neuromodulatory systems rather than reflecting neuron to neuron information exchange. If the gastric input anchors body centered representations within a single functional system, the differential regional expression of significant time lags requires more mechanistic explanation.

We have added a full paragraph on delays in the Discussion section:

"The gastric network is characterized by temporal fluctuations with delays between the gastric rhythm and brain regions.[…] The different factors may further be combined, i.e. neuromodulation might affect cerebrovascular reactivity(55)."

5) Related to this is the controversial claim is the statement that the newly discovered coupling between gastric rhythm and resting-state BOLD signal "appears to bind together distributed maps encoded in bodily coordinates into a coherent functional system (subsection “Gastric network and bodily space”)".This assertion should be discussed in more detail. First, the very idea that synchronous oscillations serve a high-level cognitive purpose as binding may not be completely proved, as the unresolved debate over binding-by-γ-synchrony shows (Merker, 2013; Ray and Maunsell, 2015). Rather than acting as an information-processing neural code of a sort, γ oscillations may simply be a neural signature of the balance between excitation and inhibition. Likewise, the EGG-BOLD synchrony found by the authors may reflect general homeostatic processes which do not play a particular role in cognition. Cerebral blood flow supplies nutrients to neurons, and such nutrients come from digestive processes. In this context, the slow-wave potentials of interstitial cells of Cajal may simply trigger food intake, given the fact that participants fasted for at least 90 minutes before the experiment (Subsection “Experimental procedure”). This could explain why the gastric network so neatly overlaps with areas mapping faces and hands in general (OP1 and OP4: subsection “The gastric network includes body maps associated with touch, action and vision”), the mouth (EBA: subsection “The gastric network includes body maps associated with touch, action and vision”), teeth, lips, tongue, and hand fingers (right SI: subsection “The gastric network includes body maps associated with touch, action and vision”), as well as with three medial wall motor preparation regions (CCZ, RCZp, and SMA: subsection “The gastric network includes body maps associated with touch, action and vision”) which become more active when subjects are asked to move their tongue or their hand (Amiez and Petrides, 2014). The authors previously excluded that variations in cerebral blood flow could be due to digestion, although they backed their position with a single 1980 study on regional blood flow in the conscious dog (Gallavan et al., 1980). Nevertheless, they should consider the opposite hypothesis, namely, that the lack of nutrients to digest may spawn a long-range synchrony between the stomach and brain areas related to food intake. Data on caloric intake prior to the experiment, blood ghrelin levels, or a self-reported hunger scale may have shed light on the issue. Thus, I recommend the authors discuss and rule out more mundane interpretations before asserting that "stomach-brain synchrony binds neural representations of the body".

We agree with the reviewer that the use of the term "binding", or any reference a computational role of synchronous oscillations, were inappropriate; they have been removed (see also points 6 and 7).

In the new subsection "What is the functional role of the gastric network", we explicitly acknowledge that any discussion on the functional role of the gastric network is speculative at this point. We now consider several possibilities: a general role in homeostasis, a more specific role in digestion, a simple cortical mapping of the stomach (a body part) on body part maps, or a more speculative interpretation in terms of coordination of bodily coordinates. Note that the title has also be modified (see minor point 2) to be more factual: "Stomach-brain synchrony reveals a novel, delayed-connectivity resting-state network".

6) The interesting, newly-discovered synchrony arises in a resting state: In a task-free environment, what is the functional advantage of having "different body-centered representations" anchored by a delayed-connectivity gastric network (subsection “The gastric network is a novel resting-state network”) with lags up to 3.3 seconds? Of note, previous "binding-by-synchrony" proposals were based on task-dependent results.

The term "binding", or any reference a computational role of synchronous oscillations, were inappropriate; they have been removed (see also points 5 and 7).

In the new subsection "What is the functional role of the gastric network" we now propose different functions for the gastric network. Some of them are relevant during cognitive rest (homeostatic regulation, digestion), others would be useful during actions such as navigation or grasping. Note that we also explicitly acknowledge the speculative nature of the discussion of the functional role of the gastric network.

7) Only 5 out of 12 nodes of the gastric network have a clear somatotopic organisation (subsection “Experimental procedure”): thus, "binding" interpretation of the authors does not explain the role that the majority of the nodes play in the network. Even the nodes which do have a somatotopic arrangement contain neurons tuned to the external perception of body parts, be it touch (nodes partially overlapping with SI and SII), motor preparation (MWM) or vision (EBA). As the authors concede, the stomach is "an organ that cannot be easily touched, moved, or seen (subsection “Gastric network and bodily space”)". It is intriguing and difficult to understand why the network revealed here seems organized around brain areas involved in body and bodily space as well as the stronger claim that it is gastric rhythm by itself that functionally links these areas.

The term "binding", or any reference a computational role of synchronous oscillations, were inappropriate; they have been removed (see also points 5 and 6).

While we retained the notion of functional connectivity between areas (as routinely used in fMRI studies), we no longer make the stronger claim that the gastric rhythm functionally links these areas.

In the new subsection "What is the functional role of the gastric network", we explicitly propose different tentative and non-mutually exclusive interpretations. The hypothesis of a simple cortical mapping of the stomach (a body part) on body part maps accounts for 5 out of the 12 nodes of the gastric network. As rightly underlined by the reviewer, this hypothesis involves a convergence between external and internal information and we know highlight the multi-sensory nature of SI, EBA and MWM.

We have also clarified the more speculative hypothesis that the gastric network coordinates areas involved in body and bodily space:

"However, the gastric network is not limited to body maps, it also comprises regions that play a role in mapping the external space in bodily coordinates, namely, the right superior parieto-occipital sulcus, dorsal precuneus and RSC. […] In this view, the function of gastric-BOLD coupling in those 9 areas would be to co-register body-centered maps of the body and of the external space. "

8) The authors make the point that SI, SII and MWM "likely" have direct gastric input, hence their claim that gastric-BOLD coupling has a neural origin (subsection “Neural origin of gastric-BOLD coupling”). Yet, the insular cortex is almost completely missing from the gastric network (subsection “Marginal gastric-brain coupling in the insula and autonomic control networks”). This is surprising, given that the insula is certainly linked to the stomach and likely supports representations of the whole body (Craig, 2002, 2009). The authors should advance a clear hypothesis to explain this seemingly counter-intuitive finding which relates to the real meaning of the brain-stomach synchronization.

We have considerably refined our analysis of the insula (subsection “Gastric-brain coupling in the right posterior insula”). Indeed, the use of statistical thresholds results in a binary output. To avoid this issue, we computed the effect size of gastric-BOLD coupling in the 6 insular ROIs, as well as in the 12 gastric nodes. Effect size in the right posterior insula was similar to that of the weakest node of the gastric network. This indicates the right posterior insula does show evidence for coupling with the stomach, provided signal-to-noise ratio is increased by averaging within a region of interest. We also analyzed the phase delay of the right posterior insula and show that the right posterior insula behaves as an early node, which is in line with its role in visceroception.

In addition, we underline in the Discussion section that the modest involvement of the insula in the present data might be due to the absence of an interoceptive task.

9) The authors should discuss the possibility that unusual measuring circumstances, wherein the participants lie in the scanner with electrodes fitted on their stomach area, create unusual conditions leading heightened awareness of the own body. This is not to say that activity in all nodes of the current network is directly and primarily related to body awareness. Indeed, some areas (SI, SII) receive direct gastric input. The network presented here may be less homogeneous than argued here and that activity in some of the nodes may have multiple bases. This would particularly seem to be the case for EBA and one can imagine different explanations for activity in EBA and for the fact that it comes on the latest. In the same vein, some arguments in defense of the representational specificity of the network appear overly speculative, e.g. suggestions about the spatial aspects of the body representation function.

The unusual measuring circumstances are now acknowledged. In the new subsection entitled "What is the functional role of the gastric network?", we specifically consider the possibility that participants' attention was drawn to their internal state (notably hunger, see also point 5).

Regarding delays, in the revised version we only highlight areas that come first as regions receiving visceral inputs. We have added a new subsection entitled "Delays and directionality of interactions", where we list the different possible explanations for long delays.

Last, we concur with the reviewer about the speculative discussion of the functional role of the gastric network. This section of the Discussion section has been completely rewritten to offer a more balanced and comprehensive view of the different possibilities, trying to synthetize all comments from the reviewers on the functional role of the gastric network (i.e., points 5, 6,7,9). The hypothesis that gastric network coordinates different body-centered maps is explained more clearly, and its speculative nature highlighted.